# Deep Ocean Temperatures through Time

Paul J Valdes[1], Christopher R Scotese[2], Daniel J Lunt[1]

[1] School of Geographical Sciences, University of Bristol, Bristol BS8 1SS, UK

[2] Northwestern University, Dept Earth & Planetary Sci, Evanston, IL USA

Correspondence to Paul Valdes (P.J.Valdes@bristol.ac.uk)

## Abstract

Benthic oxygen isotope records are commonly used as a proxy for global mean surface temperatures during the late Cretaceous and Cenozoic, and the resulting estimates have been extensively used in characterising major trends and transitions in the climate system, and for analysing past climate sensitivity. However, some fundamental assumptions governing this proxy have rarely been tested. Two key assumptions are: (a) benthic foraminiferal temperatures are geographically well-mixed and are linked to surface high latitude temperatures, and (b) surface high latitude temperatures are well correlated with global mean temperatures. To investigate the robustness of these assumptions through geological time, we performed a series of 109 climate model simulations using a unique set of paleogeographical reconstructions covering the entire Phanerozoic at the stage-level. The simulations have been run for at least 5000 model years to ensure that the deep ocean is in dynamic equilibrium. We find that the correlation between deep ocean temperatures and global mean surface temperatures is good for the Cenozoic and thus the proxy data are reliable indicators for this time period, albeit with a standard error of 2K. This uncertainty has not normally been assessed and needs to be combined with other sources of uncertainty when, for instance, estimating climate sensitivity based on using $\delta^{18}O$ measurements from benthic foraminifera. The correlation between deep and global mean surface temperature becomes weaker for pre-Cenozoic time periods (when the paleogeography is significantly different than the present-day). The reasons for the weaker correlation include variability in the source region of the deep water (varying hemispheres but also varying latitudes of sinking), the depth of ocean overturning (some extreme warm climates have relatively shallow and sluggish circulations weakening the link between surface and deep ocean), and the extent of polar amplification (e.g. ice albedo feedbacks). Deep ocean sediments prior to the Cretaceous are rare, so extending the benthic foram proxy further into deeper time is problematic, but the model results presented here would suggest that the deep ocean temperatures from such time periods would probably be an unreliable indicator of global mean surface conditions.

## 1. Introduction

One of the most widely used proxies for estimating global mean surface temperature through the last 100 million years is benthic $\delta^{18}O$ measurements from deep sea foraminifera (Zachos et al., 2001), (Zachos et al., 2008), (Cramer et al., 2009), (Friedrich et al., 2012), (Westerhold et al., 2020). Two key underlying assumptions are that $\delta^{18}O$ from benthic foraminifera represents deep ocean temperature (with a correction for ice volume and any vital effects), and that the deep ocean water masses originate from surface water in polar regions. By further assuming that polar surface temperatures are well correlated with global mean surface temperatures, then deep ocean isotopes can be assumed to track global mean surface temperatures. More specifically, (Hansen et al., 2008), and (Hansen and Sato, 2012) argue that changes in high latitude sea surface temperatures are approximately proportional to global mean surface temperatures because changes are generally amplified at high latitudes but that this is offset because temperature change is amplified over land areas. They therefore directly equate changes in benthic ocean temperatures with global mean surface temperature.

The resulting estimates of global mean surface air temperature have been used to understand past climates (e.g. (Zachos et al., 2008), (Westerhold et al., 2020)). Combined with estimates of atmospheric $CO_2$ they have also been used to estimate climate sensitivity (e.g. (Hansen et al., 2013)) and hence contribute to the important ongoing debate about the likely magnitude of future climate change.

However, some of the underlying assumptions behind the method remains largely untested, even though we know that there are major changes to paleogeography and consequent changes in ocean circulation and location of deep-water formation in the deep past (e.g. (Lunt et al., 2010; Nunes and Norris, 2006); (Donnadieu et al., 2016); (Farnsworth et al., 2019a); (Ladant et al., 2020)). Moreover, the magnitude of polar amplification is likely to vary depending on the extent of polar ice caps, and changes in cloud cover (Sagoo et al., 2013), (Zhu et al., 2019). These issues are likely to modify the correlation between deep ocean temperatures and global mean surface temperature or, at the very least, increase the uncertainty in reconstructing past global mean surface temperatures.

The aim of this paper is two-fold, (1) we wish to document the setup and initial results from a unique set of 109 climate model simulations of the whole Phanerozoic era (last 540 million years) at the stage level (approximately every 5 million years), and (2) we will use these simulations to investigate

the accuracy of the deep ocean temperature proxy in representing global mean surface
temperature.
The focus of the work is to examine the link between benthic ocean temperatures and surface
conditions. However, we evaluate the fidelity of the model by comparing the model predicted ocean
temperatures to estimates of the isotopic temperature of the deep ocean during the past 110
million years ((Zachos et al., 2008), (Cramer et al., 2009), (Friedrich et al., 2012)), and model
predicted surface temperatures to the sea surface temperatures estimates of (O'brien et al., 2017)
and (Cramwinckel et al., 2018). This gives us confidence that the model is behaving plausibly but we
emphasise that the fidelity of the simulations is strongly influenced by the accuracy of $CO_2$ estimates
through time. We then use the complete suite of climate simulations to examine changes in ocean
circulation, ice formation, and the impact on ocean and surface temperature. Our paper will not
consider any issues associated with assumptions regarding the relationship between deep-sea
foraminifera $\delta^{18}O$ and various temperature calibrations because our model does not simulate the
$\delta^{18}O$ of sea water (or vital effects).

## 2. Simulation Methodology
### 2.1 Model Description
We use a variant of the Hadley Centre model, HadCM3 ((Pope et al., 2000), (Gordon et al., 2000))
which is a coupled atmosphere-ocean-vegetation model. The specific version, HadCM3BL-M2.1aD, is
described in detail in (Valdes et al., 2017). The model has a horizontal resolution of 3.75° x 2.5° in
longitude/latitude (roughly corresponding to an average grid box size of ~300km) in both the
atmosphere and the ocean. The atmosphere has 19 unequally spaced vertical levels, and the ocean
has 20 unequally spaced vertical levels. To avoid singularity at the poles, the ocean model always has
to have land at the poles (90N and 90S), but the atmosphere model can represent the poles
correctly (i.e. in the pre-industrial geography, the atmosphere considers there is sea ice covered
ocean at the N. Pole but the ocean model has land and hence there is no ocean flow across the
pole).  Though HadCM3 is a relatively low resolution and low complexity model compared to the
current CMIP5/CMIP6 state-of-the-art model, its performance at simulating modern climate is
comparable to many CMIP5 models (Valdes et al., 2017). The performance of the dynamic
vegetation model compared to modern observations is also described in (Valdes et al., 2017) but the
modern deep ocean temperatures are not described in that paper. We therefore include a
comparison to present day observed deep ocean temperatures in section 3.1.
To perform paleo-simulations, several important modifications to the standard model described in
(Valdes et al., 2017) must be incorporated:
(a) The standard pre-industrial model uses a prescribed climatological pre-industrial ozone

concentration (i.e. prior to the development of the "ozone" hole) which is a function of

latitude, atmospheric height, and month of the year. However, we do not know what the

distribution of ozone should be in these past climates.  (Beerling et al., 2011) modelled small

changes in tropospheric ozone for the early Eocene and Cretaceous but no comprehensive

stratospheric estimates are available. Hence most paleoclimate model simulations assume

unchanging concentrations. However, there is a problem with using a prescribed ozone

distribution for paleo-simulations because it does not incorporate ozone feedbacks

associated with changes in tropospheric height.  During warm climates, the model predicts

that the tropopause would rise. In the real world, ozone would track the tropopause rise.

However, this rising ozone feedback is not included in our standard model. This leads to

substantial extra warming and artificially increases the apparent climate sensitivity.

Simulations of future climate change have shown that ozone feedbacks can lead to an over-

estimate of climate sensitivity by up to 20% ((Dietmuller et al., 2014), (Nowack et al., 2015))

(Hardiman et al., 2019). Therefore, to incorporate some aspects of this feedback, we have

changed the ozone scheme in the model. Ozone is coupled to the model predicted

tropopause height every model timestep in the following simple way:

• $2.0 \times 10^{-8}$ kg/kg in the troposphere

• $2.0 \times 10^{-7}$ kg/kg at the tropopause

• $5.5 \times 10^{-6}$ kg/kg above the tropopause

• $5.5 \times 10^{-6}$ kg/kg at the top model level.

These values are approximate averages of present-day values and were chosen so that the

tropospheric climate of the resulting pre-industrial simulation was little altered compared

with the standard preindustrial simulations; the resulting global mean surface air

temperatures differed by only 0.05 ˚C. These modifications are similar to those used in the

FAMOUS model (Smith et al., 2008) except that the values in the stratosphere are greater in

our simulation, largely because our model vertical resolution is higher than in FAMOUS.

Note that these changes improve upon the scheme used by (Lunt et al., 2016) and

(Farnsworth et al., 2019a). They used much lower values of stratospheric ozone and had no

specified value at the top of the model. This resulted in their model having ~ 1˚C cold bias

for pre-industrial temperatures and may have also affected their estimates of climate
sensitivity.
(b) The standard version of HadCM3 conserves the total volume of water throughout the
atmosphere and ocean (including in the numerical scheme) but several processes in the
model "lose or gain" water:

1. Snow accumulates over ice sheets but there is no interactive loss through iceberg
calving resulting in an excess loss of fresh water from the ocean.

2. The model caps salinity at a maximum of 45 PSU (and a minimum of 0 PSU), by
artificially adding/subtracting fresh water to the ocean.  This mostly affects small,
enclosed seas (such as the Red Sea or enclosed Arctic) where the model does not
represent the exchanges with other ocean basins.

3. Modelled river runoff includes some river basins which drain internally. These often
correspond to relatively dry regions, but any internal drainage simply disappears
from the model.

4. The land surface scheme includes evaporation from sub-grid scale lakes (which are
prescribed as a lake fraction in each grid box, at the start of the run). The model
does not represent the hydrological balance of these lakes, consequently the
volume of the lakes does not change.  This effectively means that there is a net
source/sink of water in the model in these regions.

In the standard model, these water sources/sinks are approximately balanced by a flux of
water into the surface ocean. This is prescribed at the start of the run and does not vary
during the simulations. It is normally set to a pre-calculated estimate based on an old
HadCM3-M1 simulation. The flux is strongest around Greenland and Antarctica and is
chosen such that it approximately balances the water loss described in (1) i.e. the net snow
accumulation over these ice sheets. There is an additional flux covering the rest of the
surface ocean which approximately balances the water loss from the remaining three terms
(2-4). The addition of this water flux keeps the global mean ocean salinity approximately
constant on century time scales. However, depending on the simulation, the drift in average
oceanic salinity can be as much as 1PSU per thousand years and thus can have a major
impact on ultra-long runs of >5000 years (Farnsworth et al., 2019a).
For the paleo-simulations in this paper, we therefore take a slightly different approach.
When ice sheets are present in the Cenozoic, we include the water flux (for the relevant
hemisphere) described in (1) above, based on modern values of iceberg calving fluxes for
each hemisphere. However, to ensure that salinity is conserved, we also interactively
calculate an additional globally uniform surface water flux based on relaxing the volume
mean ocean salinity to a prescribed value on a 20-year timescale. This ensures that there is
no long-term trend in ocean salinity. Tests of this update on the pre-industrial simulations
revealed no appreciable impact on the skill of the model relative to the observations. We
have not directly compared our simulations to the previous runs of the (Farnsworth et al.,
2019a) because they use different $CO_2$ and different paleogeographies. However in practice,
the increase of salinity in their simulations is well mixed and seems to have relatively little
impact on the overall climate and ocean circulation.
We have little knowledge of whether ocean salinity has changed through time, and so keep
the prescribed mean ocean salinity constant across all simulations.

2.2 Model Boundary Conditions
There are several boundary conditions that require modification through time. In this sequence of
simulations, we only modify three key time-dependent boundary conditions: 1) the solar constant, 2)
atmospheric $CO_2$ concentrations and, 3) paleogeographic reconstructions. We set the surface soil
conditions to a uniform medium loam everywhere. All other boundary conditions (such as orbital
parameters, volcanic aerosol concentrations etc.) are held constant at pre-industrial values.
The solar constant is based on (Gough, 1981) and increases linearly at an approximate rate of 11.1
$Wm^{-2}$ per 100 Ma (0.8% per 100Ma), to 1365$Wm^{-2}$ currently.  If we assume a planetary albedo of 0.3,
and a climate sensitivity of 0.8 ˚C /$Wm^{-2}$ (approximately equivalent to 3˚C per doubling of $CO_2$), then
this is equivalent to a temperature increase of ~.015˚C per million years (~8˚C over the whole of the
Phanerozoic).
Estimates of atmospheric $CO_2$ concentrations have considerable uncertainty. We, therefore, use two
alternative estimates (fig. 1a). The first uses the best fit Loess curve from (Foster et al., 2017), which
is also very similar to the newer data from (Witkowski et al., 2018). The $CO_2$ levels have considerable
short and long-term variability throughout the time period. Our second estimate removes much of
the shorter-term variability in the Foster (2017) curve.  It was developed for two reasons. Firstly, a
lot of the finer temporal structure in the Loess curve is a product of differing data density of the raw
data and does not necessarily correspond to real features. Secondly, the smoother curve was heavily
influenced by a previous (commercially confidential) sparser sequence of simulations using non-
public paleogeographic reconstructions. The resulting simulations were generally in good agreement
with terrestrial proxy datasets (Harris et al., 2017). Specifically, using commercial in confidence
paleogeographies, we have performed multiple simulations at different $CO_2$ values for several stages
across the last 440 million years and tested the resulting climate against commercial-in-confidence
proxy data (Harris et al., 2017). We then selected the $CO_2$ that best matched the data. For the
current simulations, we linearly interpolated these $CO_2$ values to every stage. The resulting $CO_2$
curve looks like a heavily smoothed version of the Foster curve and is within the (large) envelope of
$CO_2$ reconstructions. The first-order shapes of the two curves are similar, though they are very
different for some time periods (e.g. Triassic and Jurassic). In practice, both curves should be
considered an approximation to the actual evolution of $CO_2$ through time which remains uncertain.
We refer to the simulation using the second set of $CO_2$ reconstructions as the "smooth" $CO_2$
simulations, though it should be recognised that the Foster $CO_2$ curve has also been smoothed. The
Foster $CO_2$ curve extends back to only 420 Ma, so we have proposed two alternative extensions back
to 540 Ma.  Both curves increase sharply so that the combined forcing of $CO_2$ and solar constant are
approximately constant over this time period (Foster et al., 2017). The higher $CO_2$ in the Foster curve
relative to the "smooth" curve is because the initial set of simulations showed that the Cambrian
simulations were relatively cool compared to data estimates for the period (Henkes et al., 2018).
2.3 Paleogeographic Reconstructions
The 109 paleogeographic maps used in the HadleyCM3 simulations are digital representations of the
maps in the PALEOMAP Paleogeographic Atlas (Scotese, 2016); (Scotese and Wright, 2018). Table 1
lists all the time intervals that comprise the PALEOMAP Paleogeographic Atlas.  The Paleo Atlas
contains one map for nearly every stage in the Phanerozoic.  A paleogeographic map is defined as a
map that shows the ancient configuration of the ocean basins and continents, as well as important
topographic and bathymetric features such as mountains, lowlands, shallow sea, continental
shelves, and deep oceans. Paleogeographic reconstructions older than the oldest ocean floor (~Late-
Jurassic) have uniform deep ocean floor depth.
Once the paleogeography for each time interval has been mapped, this information is then
converted into a digital representation of the paleotopography and paleobathymetry. Each digital
paleogeographic model is composed of over 6 million grid cells that capture digital elevation
information at a 10 km x 10 km horizontal resolution and 40-meter vertical resolution. This
quantitative, paleo-digital elevation model, or "paleoDEM", allows us to visualize and analyse the
changing surface of the Earth through time using GIS software and other computer modelling
techniques. For use with the HadCM3L climate model, the original high-resolution elevation grid was
reduced to a ~111 km x ~111 km (1˚ x 1˚) grid.
For a detailed description of how the paleogeographic maps and paleoDEMs were produced the
reader is referred to (Scotese, 2016); (Scotese and Schettino, 2017); (Scotese and Wright, 2018).
(Scotese and Schettino, 2017) includes an annotated bibliography of the more than 100 key sources
of paleogeographic information.   Similar paleogeographic paleoDEMs have been produced by
(Baatsen et al., 2016) and (Verard et al., 2015).
The raw paleogeographic data reconstructs paleo-elevations and paleo-bathymetry at a resolution of
$1^{\circ}$ x $1^{\circ}$. These data were re-gridded to $3.75^{\circ}$ x $2.5^{\circ}$ resolution that matched the GCM using a simple
area (for land sea mask) or volume (for orography and bathymetry) conserving algorithm. The
bathymetry was lightly smoothed (using a binomial filter) to ensure that the ocean properties in the
resulting model simulations were numerically stable. The high latitudes had this filter applied
multiple times. The gridding sometimes produced single grid point enclosed ocean basins,
particularly along complicated coastlines, and these were manually removed. Similarly, important
ocean gateways were reviewed to ensure that the re-gridded coastlines preserved these structures.
The resulting global fraction of land is summarized in fig.1b and examples are shown in figure 2. The
original reconstructions can be found at https://www.earthbyte.org/paleodem-resource-scotese-
and-wright-2018/. Maps of each HadCM3L paleogeography are included in the supplementary
figures.
The paleogeographic reconstructions also include an estimate of land ice area ((Scotese and Wright,
2018); fig.1c). These were converted to GCM boundary conditions assuming a simple parabolic
shape to estimate the ice sheet height. These ice reconstructions suggest small amounts of land ice
were present during the early Cretaceous, unlike (Lunt et al., 2016) who used ice-free Cretaceous
paleogeographies.
2.4 Spin up Methodology
The oceans are the slowest evolving part of the modelled climate system and can take multiple
millennia to reach equilibrium, depending on the initial condition and climate state. To speed up the
convergence of the model, we initialized the ocean temperatures and salinity with the values from
previous model simulations from similar time periods using the commercial in confidence
paleogeographies. Specifically, we had a set of 17 simulations covering the last 440Ma. We selected
the nearest simulation to the time period. For instance, the 10.5 Ma, 14.9 Ma, and the 19.5Ma
simulations were initialised from the 13Ma simulation performed using the alternative
paleogeographies. Table 2 summarises the simulations performed in this study and shows the
initialisation of the model. The Foster $CO_2$ simulations were initialised from the end point of the
smooth $CO_2$ simulations. In the first set of simulations (smooth $CO_2$) we also attempted to accelerate
the spin up by using the ocean temperature trends at year 500 to linearly extrapolate the bottom 10
level temperatures for a further 1000 years. This had limited success and was not repeated. The
atmosphere variables were also initialized from the previous model simulations but the spin-up of
the atmosphere is much more rapid and did not require further intervention.
Simulations were run in parallel so were not initialised from the previous stage results using these
paleogeographies. In total, we performed almost 1 million years of model simulation and if we ran
simulations in sequence, it would have taken 30 years to complete the simulations. By running these
in parallel, initialised from previous modelling studies, we reduced the total run time to 3 months,
albeit using a substantial amount of our high-performance computer resources.
Although it is always possible that a different initialization procedure may produce different final
states, it is impossible to explore the possibility of hysteresis/bistability without performing many
simulations for each period, which is currently beyond our computing resources. Previous studies
using HadCM3L (not published) with alternative ocean initial states (isothermal at 0C, 8C, and 16C)
have not revealed multiple equilibria but this might have been because we did not locate the
appropriate part of parameter space that exhibits hysteresis. However, other studies have shown
such behaviour (e.g. (Baatsen et al., 2018)). This remains a caveat of our current work and which we
wish to investigate when we have sufficient computing resource.
The simulations were then run until they reached equilibrium, as defined by:
1. The globally and volume integrated annual mean ocean temperature trend is less than

$1^{\circ}$C/1000 year, in most cases considerably smaller than this. We consider the volume

integrated temperature because it includes all aspects of the ocean. However, it is

dominated by the deep ocean trends and is near identical to the trends at a depth of 2731m

(the lowest level that we have archived for the whole simulation).

2. The trends in surface air temperature are less than $0.3^{\circ}$C/1000 year.
3. The net energy balance at the top of the atmosphere, averaged over 100-year period at the

end of the simulation, is less than 0.25 $Wm^{-2}$ (in more than 80% of the simulations, the

imbalance is less than 0.1 $Wm^{-2}$). The Gregory plot (Gregory et al., 2004) implies surface

temperatures are within $0.3^{\circ}$C of the equilibrium state.

These target trends were chosen somewhat arbitrarily but are all less than typical orbital time scale
variability (e.g. temperature changes since the last deglaciation were approximately $5^{\circ}$C over 10,000
years). Most simulations were well within these criteria. 70% of simulations had residual net energy
balances at the top of the atmosphere of less than 0.1 $Wm^{-2}$, but a few simulations were slower to
reach full equilibrium. The strength of using multiple constraints is that a simulation may, by chance,
pass one or two of these criteria but were unlikely to pass all three tests. For example, all the models
that we extended failed at least two of the criteria. The resulting time series of volume integrated
global, annual mean ocean temperatures are shown in fig. 3. The supplementary figures also include
this for each simulation, as well as the trends at 2731m.
The "smooth" $CO_2$ simulations were all run for 5050 model years and satisfied the criteria. The
Foster-$CO_2$ simulations were initially run for a minimum of 2000 years (starting from the end of the
5000-year runs), at which point we reviewed the simulations relative to the convergence criteria. If
the simulations had not converged, we extended the runs for an additional 3000 years. If they had
not converged at the end of 5000 years, we extended them again for an additional 3000 years.  After
8000 years, all simulations had converged based on the convergence criteria. In general, the slowest
converging simulations corresponded to some of the warmest climates (final temperatures in figure
3b and 3c were generally warmer than in figure 3a). It cannot be guaranteed that further changes
will not occur; however, we note that the criteria and length of the simulations greatly exceed PMIP-
LGM  (Kageyama et al., 2017) and PMIP-DeepMIP (Lunt et al., 2017) protocols.

## 3. Results

### 3.1 Comparison of Deep Ocean Temperatures to Benthic Ocean Data

Before using the model to investigate the linkage of deep ocean temperatures to global mean surface temperatures, it is interesting to evaluate whether the modelled deep ocean temperatures agree with the deep ocean temperatures obtained from the isotopic studies of benthic foraminifera (Zachos et al., 2008; Friedrich et al., 2012). It is important to note that the temperatures are strongly influenced by the choice of $CO_2$, so we are not expecting complete agreement, but we simply wish to evaluate whether the model is within plausible ranges.  If the modelled temperatures were in complete disagreement with data, then it might suggest that the model was too far away from reality to allow us to adequately discuss deep ocean/surface ocean linkages. If the modelled temperatures are plausible, then it shows that we are operating within the correct climate space. A detailed comparison of modelled surface and benthic temperatures to data throughout the Phanerozoic, using multiple $CO_2$ scenarios, is the subject of a separate ongoing project.

Figure 4a compares the modelled deep ocean temperature to the foraminifera data from the Cenozoic and Cretaceous (115 Ma). The observed isotope data are converted to deep ocean temperature using the procedures described by  (Hansen et al., 2013). The modelled deep temperature shown in fig.4a (solid line) is the average temperature at the bottom level of the model, excluding depths less than 1000m (to avoid continental shelf locations which are typically not included in benthic data compilations). The observed benthic data are collected from a range of depths and are rarely at the very deepest levels (e.g. the new cores in  (Friedrich et al., 2011) are from current water depths ranging from 1899m to 3192m). Furthermore, large data compilations rarely include how the depth of a particular site changed with time, and thus effectively assume that any differences between basins and through time are entirely due to climate change and not to changes in depth. Hence throughout the rest of the paper we frequently use the modelled 2731m temperatures as a surrogate for the true benthic temperature. This is a pragmatic definition because the area of deep ocean reduces rapidly (e.g. there is typically only 50% of the globe deeper than 3300m). To evaluate whether this procedure gave a reasonable result, we also calculated the global average temperature at the model bottom, and at the model level at a depth of 2731m. The latter is shown by the dashed line in figure 4a. In general, the agreement between model bottom water temperatures and 2731m temperatures is very good. The standard deviation between model bottom water and constant depth of 2731m is 0.7˚C, and the maximum difference is 1.4˚C. Compared to the overall variability, this is a relatively small difference and shows that it is reasonable to assume that the deep ocean has weak vertical gradients.

The total change in benthic temperatures over the late Cretaceous and Cenozoic is well reproduced
by the model, with the temperatures associated with the "smooth" $CO_2$ record being particularly
good. We do not expect the model to represent sub-stage changes (100,000's of years) such as the
PETM excursion or OAEs, but we do expect that the broader temperature patterns should be
simulated.
Comparison of the two simulations illustrates how strongly $CO_2$ controls global mean temperature.
The Foster-$CO_2$ driven simulation substantially differs from the estimates of deep-sea temperature
obtained from benthic forams and is generally a poorer fit to data. The greatest mismatch between
the Foster curve and the benthic temperature curve is during the late Cretaceous and early
Paleogene. Both dips in the Foster-$CO_2$ simulations correspond to relatively low estimates of $CO_2$
concentrations. For these periods, the dominant source of $CO_2$ values is from paleosols (fig.1) and
thus we are reliant on one proxy methodology. Unfortunately, the alternative $CO_2$ reconstructions of
(Witkowski et al., 2018) have a data gap during these periods.
A second big difference between the Foster curve and the benthic temperature curve occurs during
the Cenomanian-Turonian.  This difference is similarly driven by a low estimate of $CO_2$ in the Foster-
$CO_2$ curve. These low $CO_2$ values are primarily based on stomatal density indices. As can be seen in
figure 1, stomatal indices frequently suggest $CO_2$ levels lower than estimates obtained by other
methods. The $CO_2$ estimates by (Witkowski et al., 2018) generally supports the higher levels of $CO_2$
(near to 1000 ppmv) that are suggested by the "smooth" $CO_2$ curve.
Both sets of simulations underestimate the warming during the middle Miocene. This issue has been
seen before in other models e.g. (You et al., 2009), (Knorr et al., 2011), (Krapp and Jungclaus, 2011)
(Goldner et al., 2014) (Steinthorsdottir, 2021). In order to simulate the surface warmth of the middle
Miocene (15 Ma), $CO_2$ concentrations in the range 460–580 ppmv were required, whereas the $CO_2$
reconstructions for this period (Foster et al., 2017) are generally quite low (250-400ppmv). This
problem may be either due to the climate models having too low a climate sensitivity or that the
estimates of $CO_2$ are too low (Stoll et al., 2019).
The original compilation of (Zachos et al., 2008) represented a relatively small portion of the global
ocean and the implicit  assumption was made that these results represented the entire ocean basin.
(Cramer et al., 2009) examined the data from an ocean basin perspective and suggested that these
inter-basin differences were generally small during the Late Cretaceous and early Paleogene (90Ma –
35 Ma) and the differences between ocean basins were larger during the late Paleogene and early
Neogene. Our model largely also reproduces this pattern. Figure 5 shows the ocean temperature at
2731 m during the late Cretaceous (69 Ma), the late Eocene (39 Ma) and the Oligocene (31 Ma) for
the "smooth"-$CO_2$ simulations. In the late Cretaceous, the model temperatures are almost identical
in the North Atlantic and Pacific (8˚C – 10˚C). There is warmer deep water forming in the Indian
Ocean (deep mixed layer depths, not shown), off the West coast of Australia (10˚C – 12˚C), but
otherwise the pattern is very homogeneous. This is in agreement with some paleoreconstructions
for the Cretaceous (e.g. (Murphy and Thomas, 2012)).
By the time we reach the late Eocene (39 Ma), the North Atlantic and Pacific remain very similar but
cooler deep water (6˚C – 8˚C) is now originating in the South Atlantic. The South Atlantic cool
bottom water source remains in the Oligocene, but we see a strong transition in the North Atlantic
to an essentially modern circulation with the major source of deep, cold water occurring in the high
southerly latitudes (3˚C – 5˚C) and strong gradient between the North Atlantic and Pacific.
Figure 5 also shows the modelled deep ocean temperatures for present day (Fig. 5d) compared to
the World Ocean Atlas Data (fig. 5e). It can be seen that the broad patterns are well reproduced in
the model, with good predictions of the mean temperature of the Pacific. The model is somewhat
too warm in the Atlantic itself and has a stronger plume from the Mediterranean than is shown in
the observations.

3.2 Comparison of Model Sea Surface Temperature to Proxy Data
The previous section focused on benthic temperatures, but it is also important to evaluate whether
the modelled sea surface temperatures are plausible (within the uncertainties of the $CO_2$
reconstructions). Figure 4b shows a comparison between the model simulations of sea surface
temperature and two published synthesis of proxy SST data. (O'brien et al., 2017) compiled $TEX_{86}$
and $\delta^{18}O$ for the Cretaceous, separated into tropical and high-latitude (polewards of 48°) regions.
(Cramwinckel et al., 2018) compiled early Cenozoic tropical SST data, using $Tex_{86}$, $\delta^{18}O$, Mg/Ca and
clumped isotopes. We compare these to modelled SST for the region 15°S to 15°N, and for the
average of Northern and Southern hemispheres between 47.5° and 60°. The proxy data includes
sites from all ocean basins and so we also examined the spatial variability within the model. This
spatial variability consists of changes along longitude (effectively different ocean basins) and
changes with latitude (related to the gradient between equator and pole). We therefore calculated
the average standard deviation of SST relative to the zonal mean at each latitude (this is shown by
the smaller tick marks) and the total standard deviation of SST relative to the regional average. In
practice, the equatorial values are dominated by inter-basin variations and hence the two measures
of spatial variability are almost identical. The high latitude variability has a bigger difference
between the longitudinal variations and the total variability, because the equator-to-pole
temperature gradient (i.e. the temperatures at the latitude limits of the region are a few degrees
warmer/colder than the average). The spatial variability was very similar for the "smooth"-$CO_2$ and
Foster-$CO_2$ simulations so, for clarity, on figure 4b we only show the results as error bars on the
model Foster-$CO_2$ simulations.
Overall, the comparison between model and data is generally reasonable. The modelled equatorial
temperatures largely follow the data, albeit with considerable scatter in the data. Both simulations
tend to be towards the warmest equatorial data in the early Cretaceous (Albian). These
temperatures largely come from $Tex_{86}$ data. There are many $\delta^{18}O$ based SST which are significantly
colder during this period. This data almost exclusively comes from cores 1050/1052 which are in the
Gulf of Mexico. It is possible that these data are offset due to a bias in the $\delta^{18}O$ of sea water because
of the relatively enclosed region. The Foster-$CO_2$ simulations are noticeably colder than the data at
the Cenomanian peak warmth, which is presumably related to the relatively low $CO_2$ as discussed for
the benthic temperatures. The benthic record also showed a cool (low $CO_2$) bias in the late
Cretaceous. This is not such an obvious feature of the surface temperatures. The Foster simulations
are colder than the "smooth"-$CO_2$ simulations during the late Cretaceous but there is not a strong
mismatch between model and data. Both simulations are close to the observations, though the
"smooth"-$CO_2$ simulations better matches the high-latitude data (but is slightly poorer with the
tropical data).
The biggest area of disagreement between model and data is at high latitudes in the mid-Cretaceous
warm period. In common with previous work with this model in the context of the Eocene (Lunt et
al., 2021) the model is considerably cooler than the data, with a 10-15$^{\circ}$C mismatch between models
and data. The polar sea surface temperature estimates may have a seasonal bias because
productivity is likely to be higher during the warmer summer months and, if we select the summer
season temperatures from the model, then the mismatch is slightly reduced by about 4$^{\circ}$C. The
problem of a cool high latitudes in models is seen in many model studies and there is increasing
evidence that this is related to the way that the models simulate clouds ((Kiehl and Shields, 2013);
(Sagoo et al., 2013); (Zhu et al., 2019; Upchurch et al., 2015)). Of course, in practice deep water is
formed during winter so the benthic temperatures do not suffer from a summer bias.
3.3 Correlation of Deep Ocean Temperatures to Polar Sea Surface Temperatures
The previous sections showed that that the climate model was producing a plausible reconstruction
of past ocean temperature changes, at least within the uncertainties of the $CO_2$ estimates. We now
use the HadCM3L model to investigate the links between deep ocean temperature and global mean
surface temperature.
In theory, the deep ocean temperature should be correlated with the sea surface temperature at the
location of deep-water formation which is normally assumed to be high latitude surface waters in
winter. We therefore compare deep ocean temperatures (defined as the average temperature at the
bottom of the model ocean, where the bottom must be deeper than 1000 m) with the average
winter sea surface temperature polewards of 60° (fig. 6).  Winter is defined as December, January,
and February in the northern hemisphere and June, July, and August in the southern hemisphere.
Also shown in Figure 6 is the best fit line, which has a slope of 0.40 (+/-0.05 at the 97.5% level), an $r^2$
of 0.59, and a standard error of 1.2°C. We obtained very similar results when we compared the polar
sea surface temperatures with the average temperature at 2731m instead of the true benthic
temperatures. We also compared the deep ocean temperatures to the mean polar sea surface
temperatures when the mixed layer depth exceeded 250 m (poleward of 50°). The results were
similar although the scatter was somewhat larger ($r^2$=0.48).
Overall, the relationship between deep ocean temperatures and polar sea surface temperatures is
clear (Figure 6) but there is considerable scatter around the best fit line, especially at the high end,
and the slope is less steep than perhaps would be expected (Hansen and Sato, 2012). The scatter is
less for the Cenozoic and late Cretaceous (up to 100 Ma: green and orange dots and triangles). If we
used only Cenozoic and late Cretaceous simulations, then the slope is similar (0.43) but $r^2$=0.92 and
standard error=0.47°C. This provides strong confirmation that benthic data is a robust
approximation to polar surface temperatures when the continental configuration is similar to the
present.
However, the scatter is greater for older time periods, with the largest divergence observed for the
warm periods of the Triassic and early Jurassic, particularly for the Foster $CO_2$ simulations (purple
and blue dots). Examination of climate models for these time periods reveals relatively sluggish and
shallow ocean circulation, with weak horizontal temperature gradients at depth (though salinity
gradients can still be important, (Zhou et al., 2008)). For instance, in the Ladinian stage, mid-Triassic
(~240Ma) the overturning circulation is extremely weak (Fig. 7). The maximum strength of the
northern hemisphere overturning cell is less than 10 Sv and the southern cell is less than 5 Sv. Under
these conditions, deep ocean water does not always form at polar latitudes. Examination of the
mixed layer depth (not shown) shows that during these time periods, the deepest mixed layer
depths are in the sub-tropics. In subtropics, there is very high evaporation relative to precipitation
(due to the low precipitation and high temperatures).  This produces highly saline waters that sinks
and spread out into the global ocean.
The idea that deep water may form in the tropics is in disagreement with early hypothesis (e.g.
(Emiliani, 1954)) but they were only considering the Tertiary and our model does not simulate any
low latitude deep water formation during this period. We only see significant tropical deep water
formation for earlier periods, and this has previously been suggested as a mechanism for warm
Cretaceous deep water formation (Brass et al., 1982), (Kennett and Stott, 1991). Deep water
typically forms in convective plumes. (Brass et al., 1982) showed that the depth and spreading of
these plumes is related to the buoyancy flux with the greatest flux leading to bottom water and
plumes of lesser flux leading to intermediate water. (Brass et al., 1982) suggested that this could
occur in warm conditions in the tropics, particularly if there was significant epicontinental seaways
and hypothesised that it "has been a dominant mechanism of deep-water formation in historical
times". It is caused by a strong buoyancy flux linked to strong evaporation at high temperatures.

Our computer model simulations are partly consistent with this hypothesis. The key aspect for the
model is a relatively enclosed seaway in the tropics and warm conditions. The paleogeographic
reconstructions (see supplementary figures) suggest an enclosed Tethyan-like seaway starting in the
Carboniferous and extending through to the Jurassic and early Cretaceous. However, the colder
condition of the Carboniferous prevents strong tropical buoyancy fluxes. When we get into the
Triassic and Jurassic, the warmer conditions lead to strong evaporation at low latitudes and bottom
water formation in the tropics. This also explains why we see more tropical deep water (and hence
poorer correlations between deep and polar surface temperatures in figure 6) when using the Foster
CO2 since this is generally higher (and hence warmer) than the smoothed $CO_2$ record.

An example of the formation of tropical deep water is shown in fig. 8. This shows a vertical cross-
section of temperature and salinity near the equator for the Ladinian stage, mid-Triassic (240Ma).
The salinity and temperature cross-section clearly shows high salinity warm waters sinking to the
bottom of the ocean and spreading out. This is further confirmed by the water age tracer, fig. 9. This
shows the water age (measured as time since it experienced surface conditions, see (England, 1995))
at 2731m in the model for the Permian, Triassic, Cretaceous and present day. The present-day
simulation shows that the youngest water is in the N. Atlantic and off the coast of Antarctica,
indicating that this is where the deep water is forming. By contrast, the Triassic period shows that
the youngest water is in the tropical Tethyan region and that it spreads out from there to fill the rest
of the ocean basin. There is no young water at high latitudes, confirming that the source of bottom
water is tropical only. For the Permian, although there continues to be a Tethyan-like tropical
seaway, the colder conditions mean that deep water is again forming at high latitudes only. The
Cretaceous is more complicated. It shows younger water in the high latitudes, but also shows some
young water in the Tethys which merges with the high latitude waters. Additional indicators of the
transitional nature of the Cretaceous are the mixed layer depth (see supplementary figures).  This is
a measure of where water is mixing to deeper levels.  For this time period, there are regions of deep
mixed layer in both the tropics and high latitudes, whereas it is only deep in the tropics for the
Triassic and at high latitudes for present day.

This mechanism for warm deep water formation has also been seen in other climate models (e.g.
(Barron and Peterson, 1990)). However, (Poulsen et al., 2001) conclude that in his model of the
Cretaceous high-latitudes sources of deep water diminish with elevated $CO_2$ concentrations but did
not see the dominance of tropical sources. Other models (e.g. (Ladant et al., 2020)) do not show any
significant tropical deep-water formation, suggesting that this feature is potentially a model-
dependent result.
The correlation between deep ocean temperatures and the temperature of polar surface waters
differs between the "smooth" $CO_2$ simulations and the Foster $CO_2$ simulations. The slope is only 0.30
($r^2$=0.57) for the "smooth" $CO_2$ simulations whereas the slope is 0.48 ($r^2$=0.65) for the Foster
simulations. This is because $CO_2$ is a strong forcing agent that influences both the surface and deep
ocean temperatures. By contrast, if the $CO_2$ does not vary as much, then the temperature does not
vary as much, and the influence of paleogeography becomes more important. These
paleogeographic changes generally cause subtle and complicated changes in ocean circulation that
affect the location and latitude of deep-water formation.
In contrast, the mid-Cretaceous is also very warm but the continental configuration (specifically, land
at high southern latitudes) favours the formation of cool, high latitude deep water. Throughout the
Cretaceous there is significant southern high latitude source of deep water and hence deep-water
temperatures are well correlated with surface high latitude temperatures. The strength of this
connection, however, may be over exaggerated in the model. Like many climate models, HadCM3
underestimates the reduction in the pole-to-Equator sea surface temperature (Lunt et al., 2012),
(Lunt et al., 2021). This means that during the Cretaceous the high latitudes are probably too cold.
Consequently, some seasonal sea ice does form which encourages the formation of cold deep-water,
via brine rejection.
In the late Eocene (~40 Ma), the ocean circulation is similar to the Cretaceous, but the strong
southern overturning cell is closer to the South Pole, indicating that the main source of deep water
has moved further polewards. The poleward movement of the region of downwelling waters
explains some of the variability between deep ocean temperatures and temperature of polar surface
waters.
For reference, we also include the present-day meridional circulation. The modern southern
hemisphere circulation is essentially a strengthening of late Eocene meridional circulation. The
Northern hemisphere is dominated by the Atlantic meridional overturning circulation. The Atlantic
circulation pattern does not resemble the modern pattern of circulation until the Miocene.
3.4 Surface Polar Amplification

The conceptual model used to connect benthic ocean temperatures to global mean surface
temperatures assumes that there is a constant relationship between high latitude sea surface
temperatures and global mean annual mean surface air temperature.  (Hansen and Sato, 2012)
argue that this amplification is partly related to ice-albedo feedback but also includes a factor
related to the contrasting amplification of temperatures on land compared to the ocean. To
investigate the stability of this relationship, fig. 10 shows the correlation between polar winter sea
surface temperatures ($60^o$ - $90^o$) and global mean surface air temperature.  The polar temperatures
are the average of the two winter hemispheres (i.e. average of DJF polar SSTs in the Northern
hemisphere and JJA polar SSTs in the Southern hemisphere). Also shown is a simple linear
regression, with an average slope of 1.3 and with an $r^2$ = 0.79. If we only use Northern polar winter
temperatures, the slope is 1.1; if we only use Southern polar winter temperatures, then the slope is
0.7. Taken separately, the scatter about the mean is considerably larger ($r^2$ of 0.5 and 0.6
respectively) than the scatter if both data sets are combined ($r^2$ = 0.79).   The difference between the
southern and northern hemisphere response complicates the interpretation of the proxies and leads
to potentially substantial uncertainties.
As expected, there appears to be a strong non-linear component to the correlation. There are two
separate regimes: 1) one with a steeper slope during colder periods (average polar winter
temperature less than about $1^o$C), and 2) a shallower slope for warmer conditions. This is strongly
linked to the extent of sea-ice cover. Cooler periods promote the growth of sea-ice which
strengthens the ice-albedo feedback mechanism resulting in a steeper overall temperature gradient
(strong polar amplification). Of course, the ocean sea surface temperatures are constrained to be -
$2^o$C but an expansion of sea ice moves this further equatorward. Conversely, the warmer conditions
result in less sea ice and hence a weaker sea ice-albedo feedback resulting in a weaker temperature
gradient (reduced polar amplification). This suggests that using a simple linear relationship (as in
(Hansen et al., 2008)) could be improved upon.
Examining the Foster $CO_2$ and "smooth" $CO_2$ simulations reveals an additional factor. If we examine
the "smooth" $CO_2$ simulations only, then the best fit linear slope is slightly less than the average
slope (1.1 vs 1.3). This can be explained by the fact that we have fewer very cold climates
(particularly in the Carboniferous) due to the relatively elevated levels of $CO_2$. However, the scatter
in the "smooth" $CO_2$ correlation is much larger, with an $r^2$ of only 0.66. By comparison, correlation
between global mean surface temperature and polar sea surface temperature using the Foster $CO_2$
has a similar overall slope to the combined set and a smaller amount of scatter. This suggests that
$CO_2$ forcing and polar amplification response have an important impact on the relationship between
global and polar temperatures. The variations of carbon dioxide in the Foster set of simulations are
large and they drive large changes in global mean temperature. Conversely significant sea-ice albedo
feedbacks characterize times when the polar amplification is important. There are several well
studied processes that lead to such changes, including albedo effects from changing ice but also
from poleward heat transport changes, cloud cover, and latent heat effects ((Sutton et al., 2007;
Alexeev et al., 2005; Holland and Bitz, 2003)). By contrast, the "smooth" $CO_2$ simulations have
considerably less forcing due to $CO_2$ variability which leads to a larger paleogeographic effect. For
instance, when there is more land at the poles, there will be more evaporation over the land areas
and hence simple surface energy balance arguments would suggest different temperatures ((Sutton
et al., 2007)) .
In figure 10, there are a few data points which are complete outliers. These correspond to
simulations in the Ordovician; the outliers happen irrespective of the $CO_2$ model that is used.
Inspection of these simulations shows that the cause for this discrepancy is related to two factors:
1) a continental configuration with almost no land in the Northern hemisphere and, 2) a
reconstruction which includes significant southern hemisphere ice cover (see fig.1 and fig 2).
Combined, these factors produced a temperature structure which is highly non-symmetric, with the
Southern high latitudes being more than $20^{\circ}C$ colder than the Northern high latitudes. This anomaly
biases the average polar temperatures shown in figure 10.
3.5 Deep Ocean Temperature versus Global Mean Temperature

The relationships described above help to understand the overall relationship between deep ocean
temperatures and global mean temperature. Figure 11 shows the correlation between modelled
deep ocean temperatures (> 1000 m) and global mean surface air temperature, and figure 12 shows
a comparison of changes in modelled deep ocean temperature compared to model global mean
temperature throughout the Phanerozoic.
The overall slope is 0.64 (0.59 to 0.69) with an $r^2$ = 0.74. If we consider the last 115 Ma (for which
exists compiled benthic temperatures), then the slope is slightly steeper (0.67 with an $r^2$ = 0.90).
Similarly, the "smooth"-$CO_2$ and the Foster-$CO_2$ simulation results have very different slopes. The
"smooth"-$CO_2$ simulations have a slope of 0.47, whereas the Foster-$CO_2$ simulations have a slope of
0.76. The root mean square departure from the regression line in figure 11 is 1.3°C. Although we
could have used a non-linear fit as we might expect such a relationship if the pole-to-equator
temperature gradient changes, all use of benthic temperatures as a global mean surface
temperature proxy are based on linear relationship.

The relatively good correlations in the fig.11 are confirmed when examining fig.12a and 12b. On
average, the deep ocean temperatures tend to underestimate the global mean change (fig.12b)
which is consistent with the regression slope being less than 1. However, the errors are substantial
with largest errors occurring during the pre-Cretaceous and can be 4-6 °C. This is an appreciable
error that would have a substantial impact on estimates of climate sensitivity. Even within the late
Cretaceous and Cenozoic, the errors can exceed 2°C which can exceed 40% of the total change.
The characteristics of the plots can best be understood in terms of figures (6 and 10). For instance,
most of the Carboniferous simulations plot below the regression line because the polar SSTs are not
well-correlated with the global mean temperature (figure 10). By contrast, the Triassic and Jurassic
Foster $CO_2$ simulations plot above the regression line because the deep ocean temperature is not
well-correlated with the polar temperatures (figure 6).
4. Discussion and Conclusion
The paper has presented the results from two unique sets of paleoclimate simulations covering the
Phanerozoic. The focus of the paper has been to use the HadCM3L climate model to evaluate how
well we can predict global mean surface temperatures from benthic foram data. This is an important
consideration because benthic microfossil data are one of the few datasets used to directly estimate
past global mean temperatures. Other methods, such as using planktonic foraminiferal estimates,
are more challenging because the sample sites are geographically sparse, so it is difficult to
accurately estimate the global mean temperature from highly variable and widely dispersed data.
This is particularly an issue for older time periods when fewer isotopic measurements from
planktonic microfossils are available and can result in a bias because most of the isotopic
temperature sample localities are from tropical latitudes (30˚S – 30˚N) (Song et al., 2019).
By contrast, deep ocean temperatures are more spatially uniform. Hence. benthic foram data has
frequently been used to estimate past global mean temperatures and climate sensitivity (Hansen et
al., 2013). Estimates of uncertainty for deep ocean temperatures incorporate uncertainties from $CO_2$
and from the conversion of $\delta^{18}O$ measurements to temperature but have not been able to assess
assumptions about the source regions for deep ocean waters and the importance polar
amplification. Of course, in practice, lack of ocean sea floor means that benthic compilations exist
only for the last 110Ma.
Changes in heat transport also play a potentially important role in polar amplification. In the
supplementary figure, we show the change in atmosphere and ocean poleward heat fluxes for each
time period. Examination of the modelled poleward heat transport by the atmosphere and ocean
shows a very complicated pattern, with all time periods showing the presence of some Bjerknes
compensation (Bjerknes, 1964) (see (Outten et al., 2018) for example in CMIP5 models). Bjerknes
compensation is where the change in ocean transport is largely balanced by an equal but opposite
change in atmospheric transport. For instance, compared to present day, the mid-Cretaceous and
Early Eocene warm simulations shows a large increase in northward atmospheric heat transport,
linked with enhanced latent heat transport associated with the warmer, moister atmosphere.
However, this is partly cancelled by an equal but opposite change in the ocean transport. E.g.
compared to present day, the early Eocene northern hemisphere atmospheric heat transport
increases by up to 0.5PW, but the ocean   transport is reduced by an equal amount. The net
transport from equator to the N. Pole changes by less than 0.1PW (i.e. less than 2% of total). Further
back in time, the compensation is still apparent, but the changes are more complicated, especially
when the continents are largely in the Southern hemisphere. Understanding the causes of these
transport changes will be the subject of another paper.
We have shown that although the expected correlation between benthic temperatures and high-
latitude surface temperatures exists, the correlation has considerable scatter. This is caused by
several factors. Changing paleogeographies results in changing locations for deep water formation.
Some paleogeographies result in significant deep-water formation in the Northern hemisphere (e.g.
our present-day configuration) although for most of the Phanerozoic, the dominant source of deep-
water formation has been southern hemisphere. Similarly, even when deep water is formed in just
one hemisphere, there can be substantial regional and latitudinal variations in its location and the
corresponding temperatures. Finally, during times of very warm climates (e.g. mid-Cretaceous) the
overturning circulation can be very weak and there is a marked decoupling between the surface
waters and deep ocean. In the HadCM3 model during hothouse time periods, high temperatures and
high rates of evaporation produce hot and saline surface waters which sink to become intermediate
and deep waters at low latitudes.
Similar arguments can be made regarding the link between global mean temperature and the
temperature at high latitudes. Particularly important is the area of land at the poles and the extent
of sea ice/land ice. Colder climates and paleogeographic configurations with more land at the pole
will result in a steeper latitudinal temperature gradient and hence exhibit a changing relationship
between polar and global temperatures.  But the fraction of land versus ocean is also important.
Finally, the overall relationship between deep ocean temperatures and global mean temperature is
shown to be relatively linear, but the slope is quite variable. In the model simulations using the
"smooth" $CO_2$ curve, the slope is substantially shallower (0.48) than slope obtained using the Foster
$CO_2$ curve (0.76). This is related to the different controls that $CO_2$ and paleogeography exert (as
discussed above). In the simulation that uses the "smooth" $CO_2$ data set, the levels of $CO_2$ do not
vary much, so the paleogeographic controls are more pronounced.
This raises the interesting conundrum that when trying to use reconstructed deep ocean
temperatures and $CO_2$ to estimate climate sensitivity, the interpreted global mean temperature also
depends, in part, on the $CO_2$ concentrations. However, if we simply use the combined slope, then
the root mean square error is approximately 1.4˚C, and the maximum error is over 4ºC. The root
mean square error is a relatively small compared to the overall changes and hence the resulting
uncertainty in climate sensitivity associated with this error is relatively small (~15%) and the $CO_2$
uncertainty dominates. However, the maximum error is potentially more significant.
Our work has not addressed other sources of uncertainty. In particular, it would be valuable to use a
water isotope-enabled climate model to better address the uncertainties associated with the
conversion of the observed benthic $\delta^{18}O$ to temperature. This requires assumptions about the $\delta^{18}O$
of sea water. We hope to perform such simulation in future work, though this is a particularly
challenging computational problem because the isotope enabled model is significantly slower and
the completion of the multi-millennial simulations required for deep ocean estimates would take
more than 18 months to complete.
Our simulations extend and develop those published by (Lunt et al., 2016), and (Farnsworth et al.,
2019b; Farnsworth et al., 2019a). The simulations reported in this paper used the same climate
model (HadCM3L) but used an improved ozone concentration and corrected a salinity drift that can
lead to substantial changes over the duration of the simulation. Our simulations also use an
alternative set of geographic reconstructions that cover a larger time period (540 Ma – Modern).
They also include realistic land ice cover estimates, which were not included in the original
simulations (except for the late Cenozoic) but generally have a small impact in the Mesozoic.
Similarly, the new simulations use two alternative models for past atmospheric $CO_2$ use more
realistic variations in $CO_2$ through time (compared with idealised constant values in Farnsworth et al
and Lunt et al), while at the same time recognizing the levels of uncertainty. Although the Foster $CO_2$
curve is more directly constrained by $CO_2$ data, it should be noted that this data come from multiple
proxies and there are large gaps in the data set. There is evidence that the different proxies have
different biases, and it is not obvious that the correct approach is to simply fit a Loess-type curve to
the $CO_2$ data. This is exemplified by the Maastrichtian. The Foster Loess curve shows a minimum in
$CO_2$ during the Maastrichtian which results in the modelled deep ocean temperatures being much
too cold. However, detailed examination of the $CO_2$ data shows most of the Maastrichtian data is
based on stomatal index reconstructions which often are lower than other proxies. Thus, the
Maastrichtian low $CO_2$, relative to other periods, is potentially driven by changing the proxy rather
than by real temporal changes.
Though the alternative, "smooth" $CO_2$ curve is not the optimum fit to the data, it does pass through
the cloud of individual $CO_2$ reconstructions and hence represents one possible "reality". For the Late
Cretaceous and Cenozoic, the "smooth" $CO_2$ simulation set does a significantly better job simulating
the deep ocean temperatures of the Friedrich/Cramer/Zachos curve.
Although the focus of the paper has been the evaluation of the modelled relationship between
benthic and surface temperatures, the simulations are a potentially valuable resource for future
studies. This includes using the simulations for paleoclimate/climate dynamic studies and for climate
impact studies, such as ecological niche modelling. We have therefore made available on our
website the results from our simulations
https://www.paleo.bristol.ac.uk/ummodel/scripts/papers/Valdes_et_al_2021.html
Data Availability
All simulation data is available from:
https://www.paleo.bristol.ac.uk/ummodel/scripts/papers/Valdes_et_al_2021.html
Author contributions
Study was developed by all authors. All model simulations were performed by PJV who also
prepared the manuscript with contributions from all co-authors.
Competing interests
The authors declare that they have no conflict of interest.


## Acknowledgments.

DJL and PJV acknowledge funding from NERC through NE/P013805/1. The production of paleogeographic digital elevation models was funded by the sponsors of the PALEOMAP Project. This work is part of the PhanTASTIC project led by Scott Wing and Brian Huber from the Smithsonian Institution's National Museum of Natural History and was initiated at a workshop supported by Roland and Debra Sauermann. This work was carried out using the computational facilities of the Advanced Computing Research Centre, University of Bristol (http://www.bris.ac.uk/acrc/). The authors declare that they have no competing interests. Data and materials availability: All data needed to evaluate the conclusions in the paper are present in the paper. Model data can be accessed at www.bridge.bris.ac.uk/resources/simulations.







**Figure 1.** Summary of boundary condition changes to model of the Phanerozoic, (a) $CO_2$
reconstructions (from Foster et al. 2017) and the two scenarios used in the models, (b) Land-sea
fraction from the paleogeographic reconstructions, and (c) land ice area input into model. The
paleogeographic reconstructions can be accessed at https://www.earthbyte.org/paleodem-
resource-scotese-and-wright-2018/. An animation of the  high-resolution (1° x 1°) and model
resolution (3.75° longitude x 2.5° latitude) maps can be found here:
https://www.paleo.bristol.ac.uk/~ggpjv/scotese/scotese_raw_moll.normal_scotese_moll.normal.ht
ml

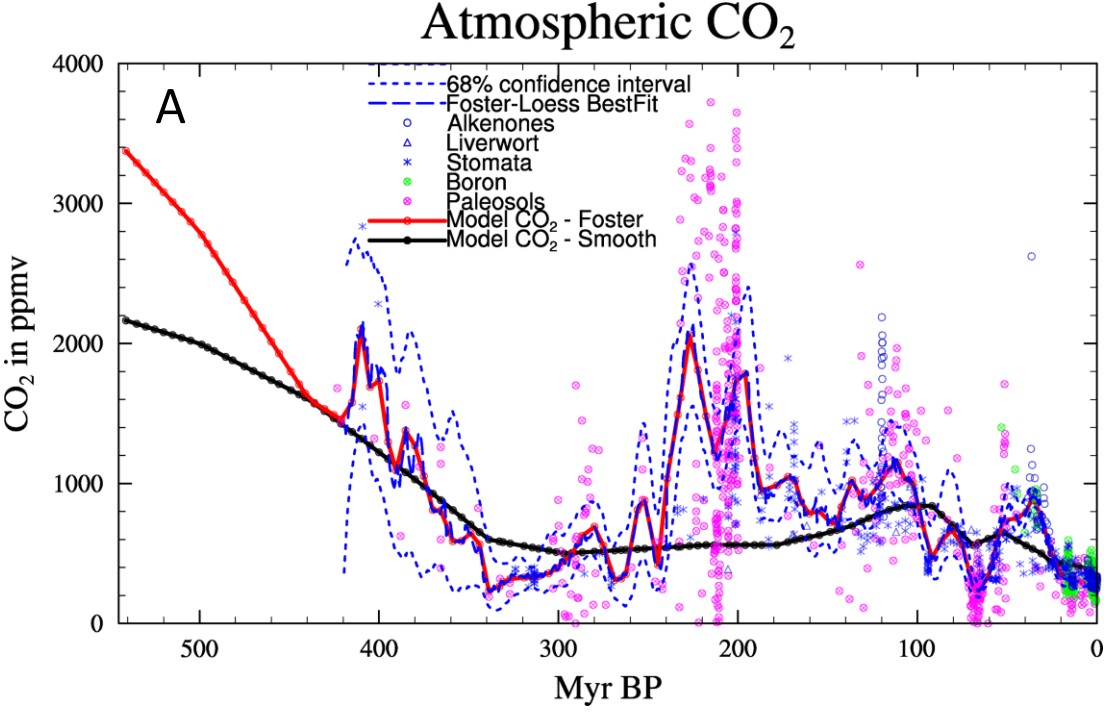


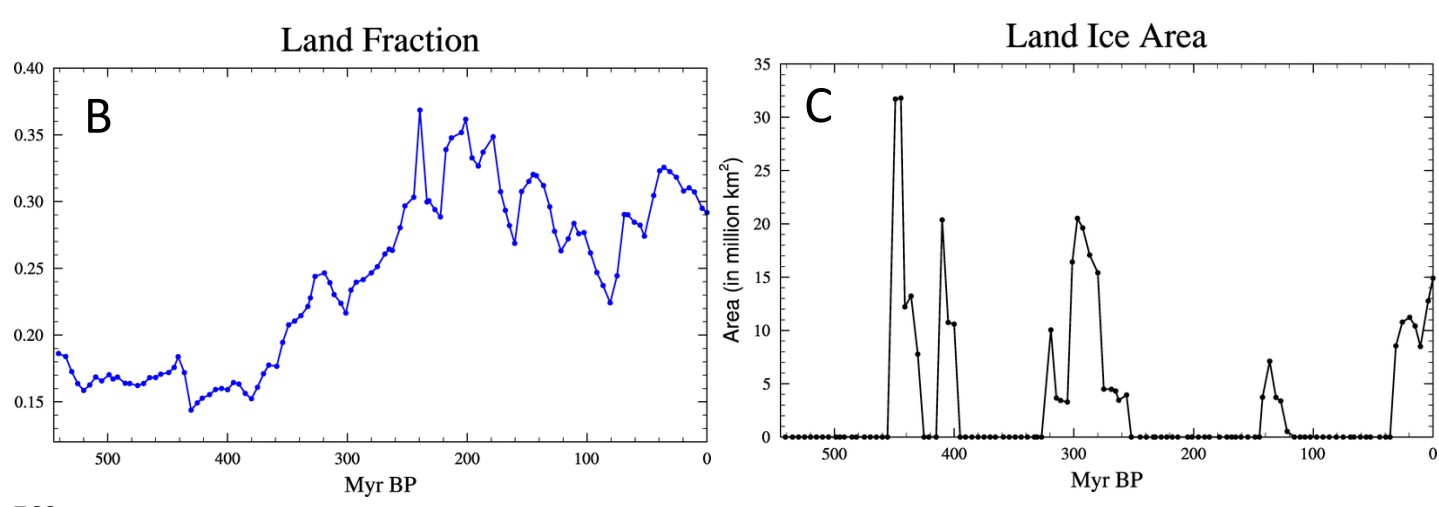



**Figure 2.** A few example paleogeographies, once they have been re-gridded onto the HadCM3L grid.
The examples are for (a) present day, (b) Albian, 102.6Ma (Lower Cretaceous), (c) Hettangian,
201.3Ma (lower Jurassic), (d) Moscovian, 311.1Ma (Pennsylvanian, Carboniferous), (e) Katian,
449.1Ma (Upper Ordovician), and (f) Fortunian, 541.0Ma (Cambrian). The top color legend refers to
the height of the ice sheets (if they exist), the middle color legend refers to heights on land (except
ice), and the lower color legend refers to the ocean bathymetry. All units are meters.

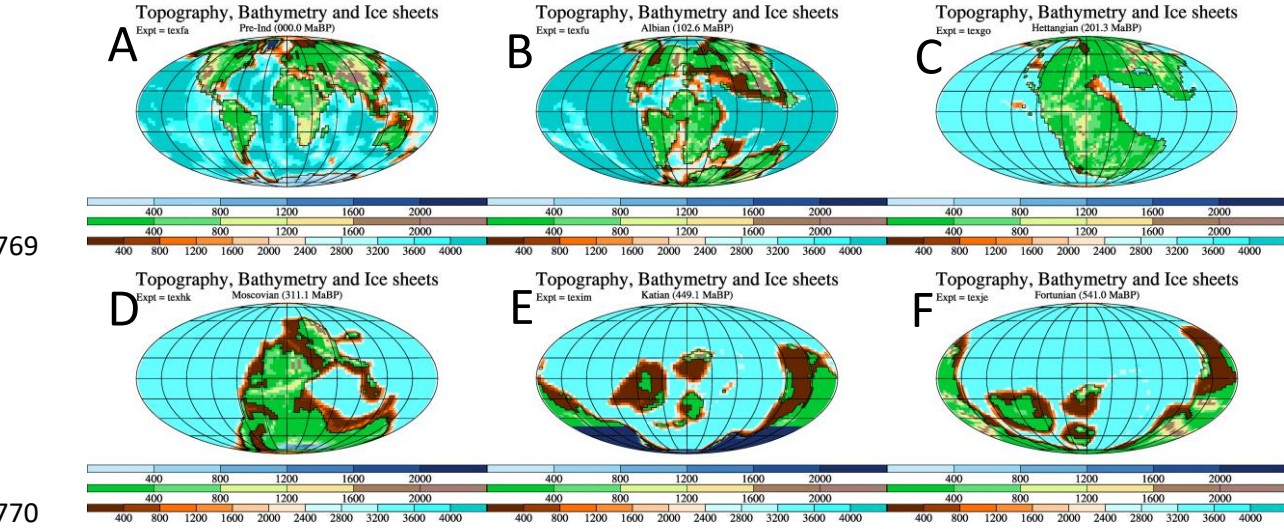












**Figure 3.** Time series of the annual, volume mean ocean temperature for all 109 simulations. (a)
shows those simulations for which 2000 years was sufficient to satisfy the convergence criteria
described in text (these were for all simulations listed in table 1 except those listed in (b) and (c)), (b)
those simulation which required 5000 years (these were for all the simulations for 31.0, 35.9, 39.5,

784     55.8, 60.6, 66.0, 69.0, 102.6, 107.0, 121.8, 127.2, 154.7, 160.4, 168.2, 172.2, 178.4, 186.8, 190.8,

196.0, 201.3, 204.9, 213.2, 217.8, 222.4, 227.0, 232.0, and 233.6 Ma BP), and (c) those simulation
which required 8000 years (these were simulations for 44.5, 52.2, 86.7, 91.9, 97.2, 111.0, 115.8,
131.2, 136.4, 142.4, 145.0, 148.6, 164.8, and 239.5 Ma BP).  The different coloured lines show the
different runs. The plot simply show the extent to which all runs have reached steady state. For
more details about specific simulations, please see the supplementary figures.

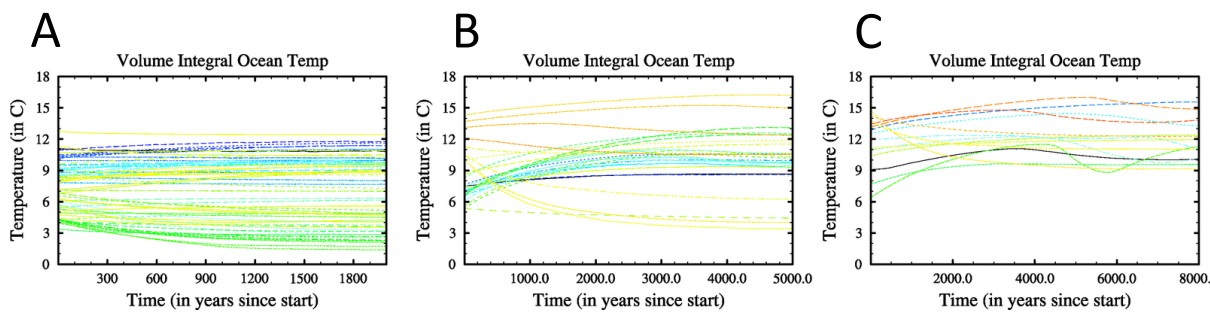




**Figure 4.** (a) Comparison of modelled deep ocean temperatures versus those from (Zachos et al.,
2008) and (Friedrich et al., 2012) converted to temperature using the formulation in (Hansen et al.,
2013). The model temperatures are global averages over the bottom layer of the model but excludes
shallow marine settings (less than 1000m). The dashed lines show the modelled global average
ocean temperatures at the model layer centered at 2731m, and (b) Comparison of modelled sea
surface temperatures with the compilations of (O'brien et al., 2017) and (Cramwinckel et al., 2018).
The data is a combination of $Tex_{86}$, $\delta^{18}O$, Mg/Ca, and clumped Isotope data. The model data shows
low latitude temperatures (averaged from 10S to 10N) and high latitude temperatures (averaged
over 47.5N to 65N and 47.5S to 65S). The Foster-$CO_2$ simulations also show a measure of the spatial
variability. The large bars show the spatial standard deviation across the whole region, and the
smaller bars shows the average spatial standard deviation along longitudes within the region. Note
that the ranges of both the x and y-axis differ between (a) and (b).

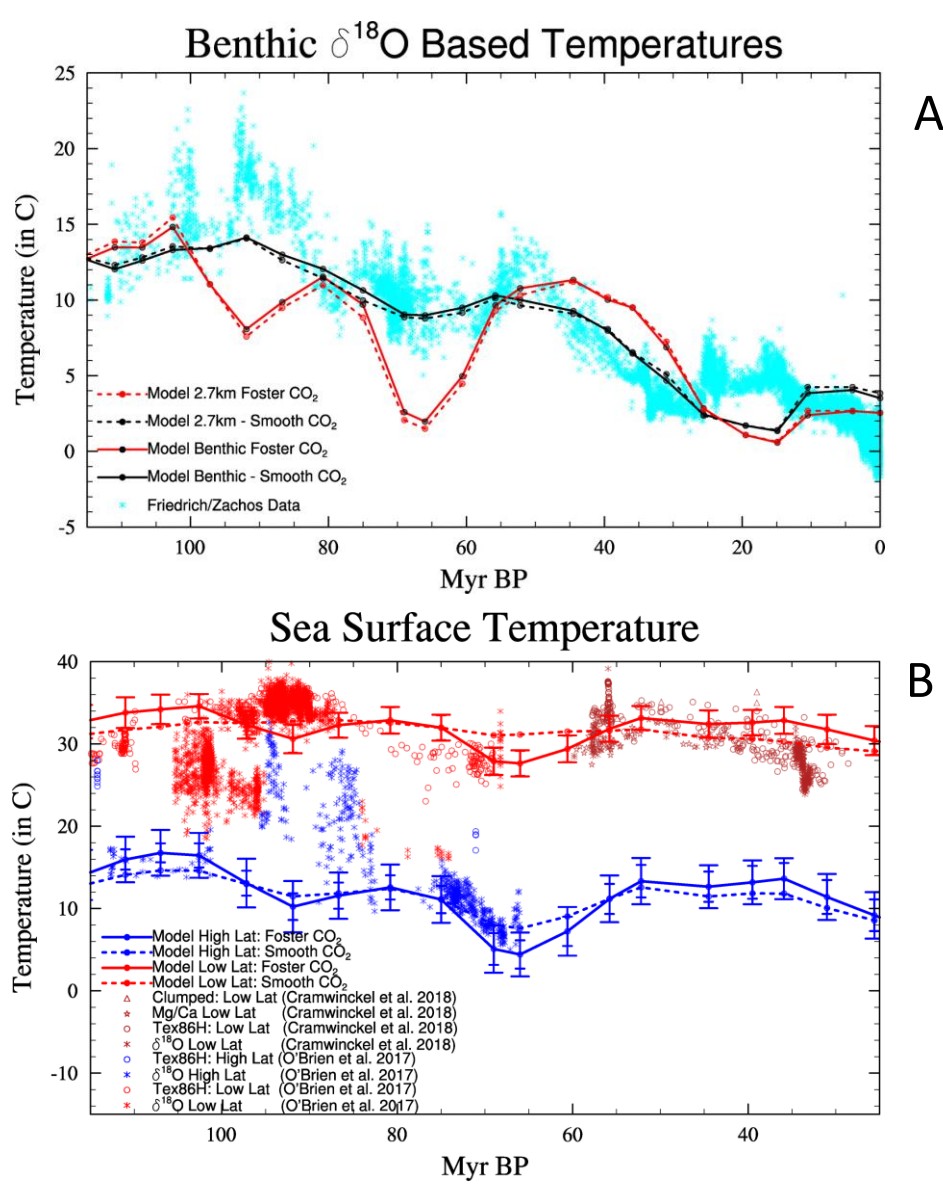

**Figure 5.** Modelled annual mean ocean temperatures are 2731m depth for three example past time
periods. The left figure is for the late Cretaceous, the center for the late Eocene (39.5Ma), and the
right for the Oligocene (31Ma). These are results from the smooth-$CO_2$ set of simulations which
agree better with the observed benthic temperature data. Also included are the pre-industrial
simulation and World Ocean Atlas 1994 observational data, provided by the NOAA-ESRL Physical
Sciences Laboratory, Boulder Colorado from their web site at https://psl.noaa.gov/. The thin black
lines show the coastlines, and the grey areas are showing where the ocean is shallower than 2731m.

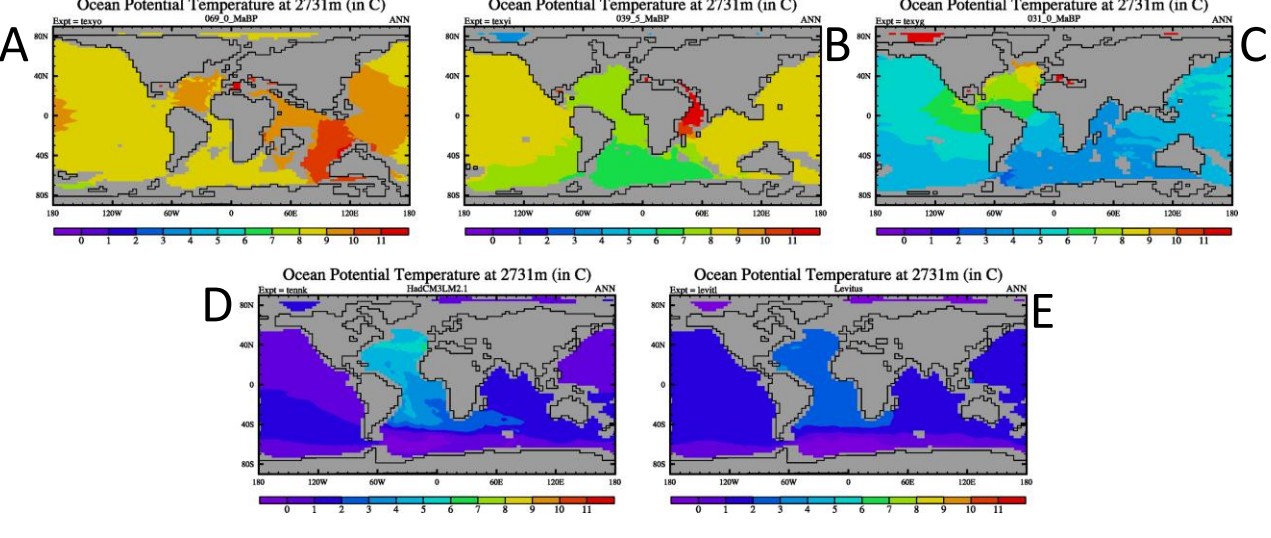



**Figure 6.** Correlations between deep ocean temperatures and surface polar sea surface
temperatures. The deep ocean temperatures are defined as the average temperature at the bottom
of the model ocean, where the bottom must be deeper than 1000m. The polar sea surface
temperatures are the average winter (i.e. northern polar in DJF and southern polar in JJA) sea
surface temperature polewards of 60°. The inverted triangles show the results from the smooth $CO_2$
simulations and the dots refer to the Foster $CO_2$ simulations. The colors refer to different geological
era.

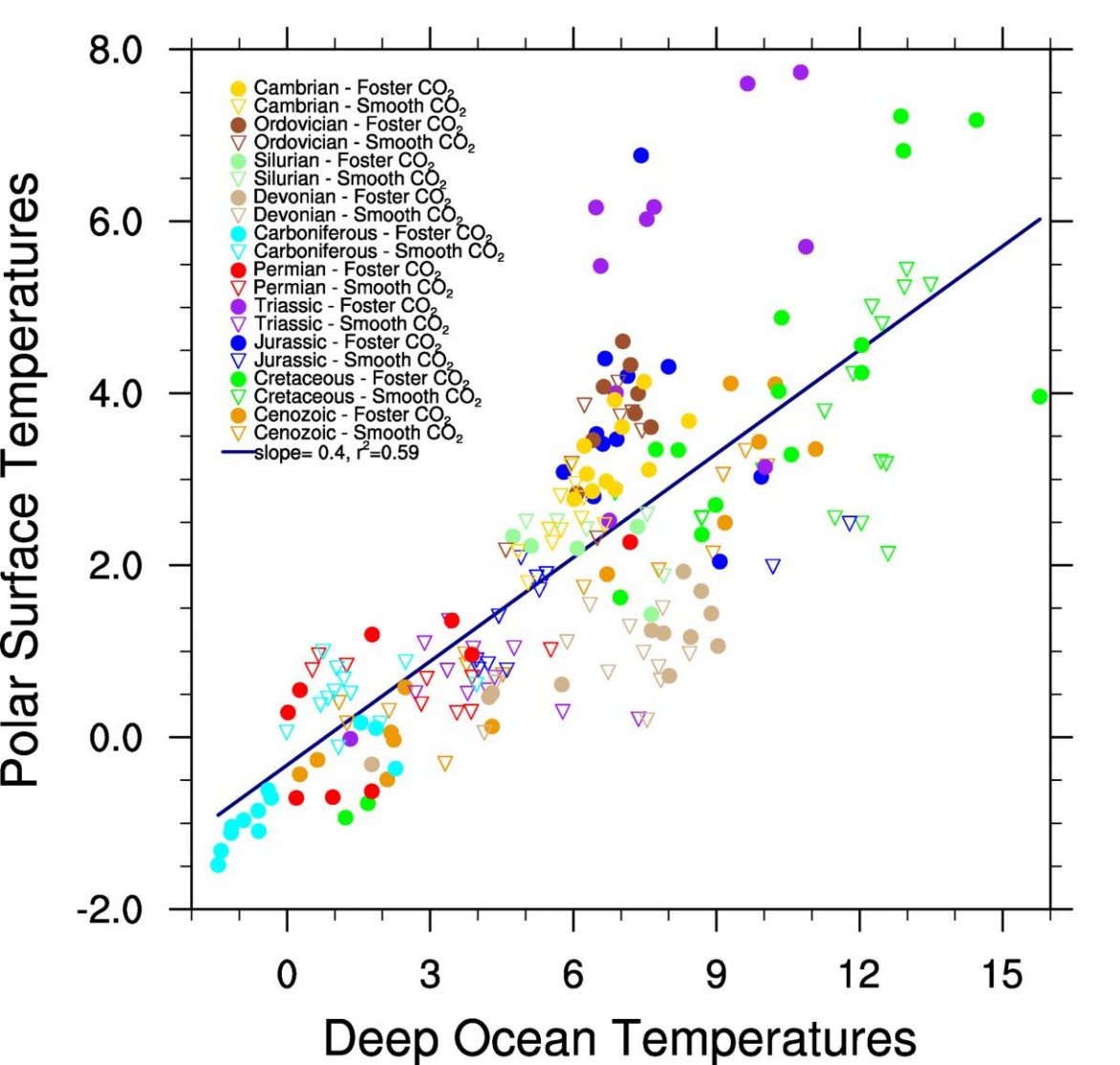


**Figure 7.** Global Ocean overturning circulation (in Sverdrup) for four different time periods for the
Foster-CO₂ simulations. Positive (yellow/red) values correspond to a clockwise circulation, negative
(dark blue/purple) values represent an anti-clockwise circulation. (a) Middle Triassic, Ladinian,
239.5Ma, (b) Lower Cretaceous, Aptian, 121.8 Ma, (c) Late Eocene, Bartonian, 39.5Ma, and (d)
Present Day. Paleogeographic reconstructions older than the oldest ocean floor (~Late-Jurassic) have
uniform deep ocean floor depth.

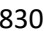



**Figure 8.** Longitudinal cross section at 20S of (a) ocean potential temperature and (b) salinity for the
Ladanian (240Ma). Temperature is in C and salinity is in PSU.

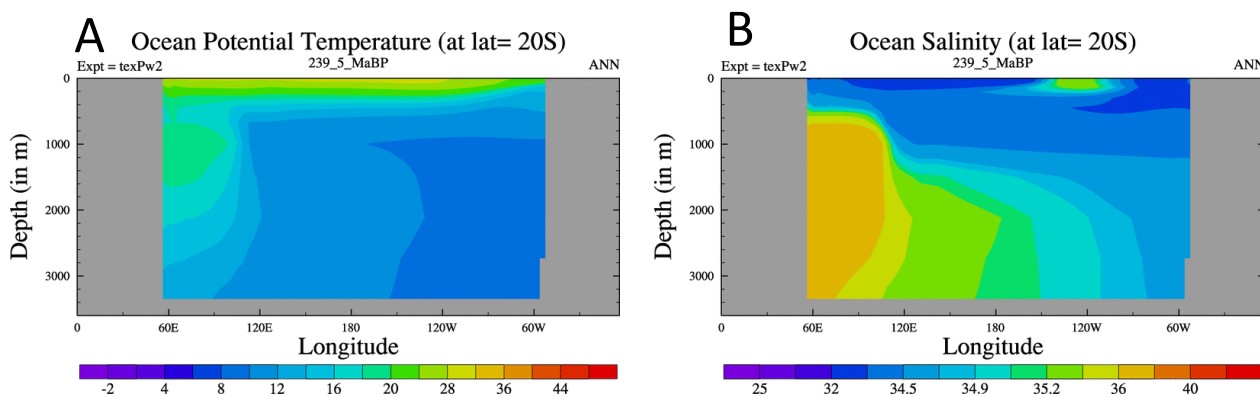


**Figure 9.** Modelled age of water tracer at 2731m for 4 different time periods (a) 265Ma, (b) 240Ma,
(c) 107Ma, and (d) 0Ma. Units are years.

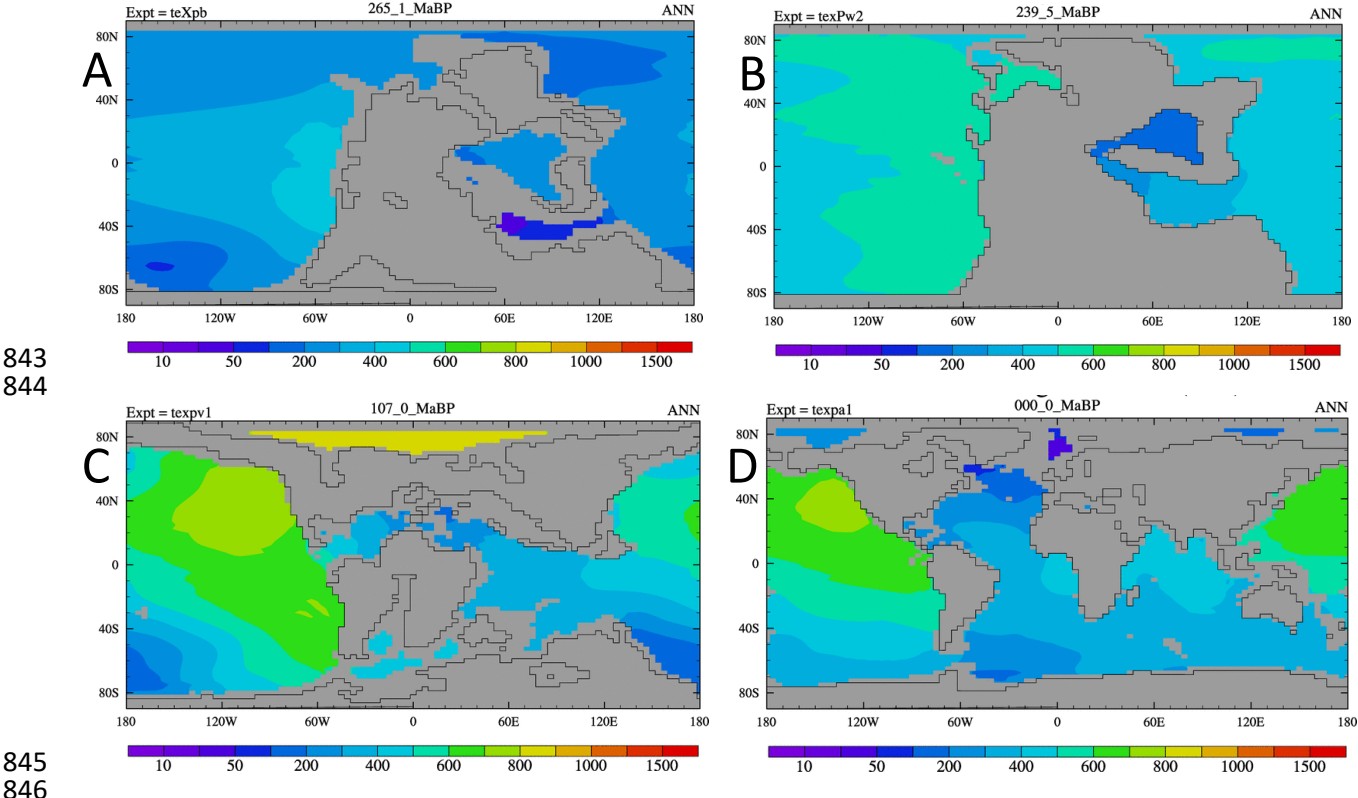



**Figure 10**. Correlation between high latitude ocean temperatures (polewards of 60°) and the annual
mean, global mean surface air temperature. The polar temperatures are the average of the two
winter hemispheres (i.e. northern DJF and southern JJA). Other details as in figure 6.


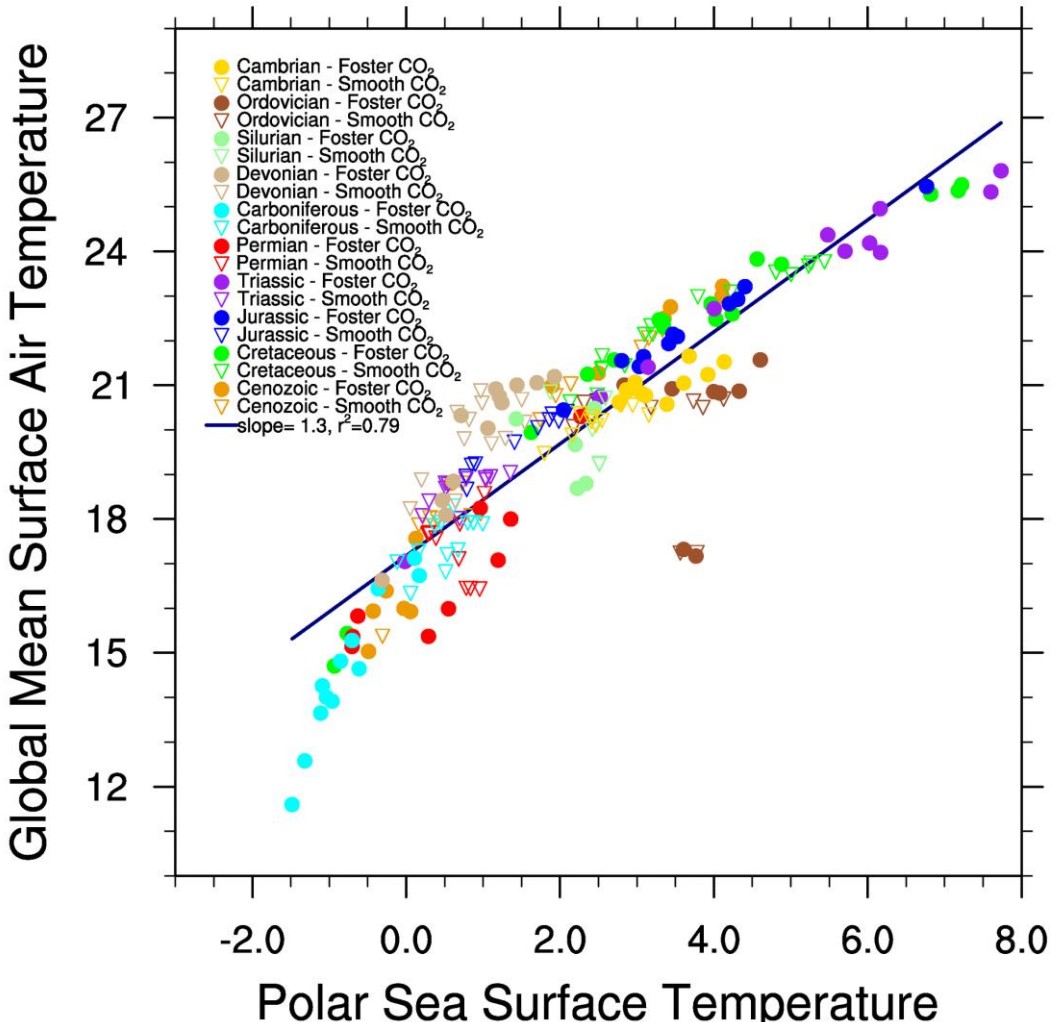


**Figure 11.** Correlation between the global mean, annual mean surface air temperature and the deep
ocean temperature. The deep ocean temperatures are defined as the average temperature at the
bottom of the model ocean, where the bottom must be deeper than 1000m. Other details as in
figure 6.

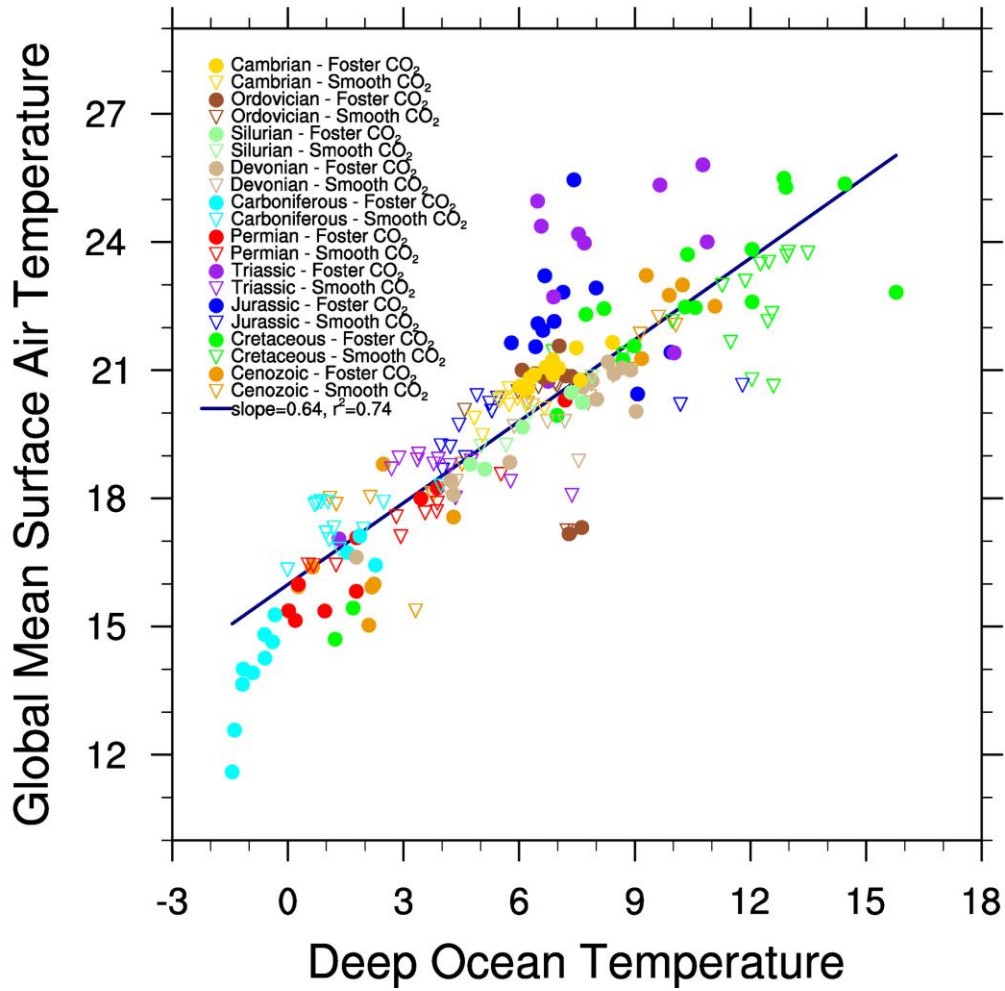



**Figure 12.** Phanerozoic Time series of modelled temperature change (relative to pre-Industrial) for
the smooth (green lines) and Foster-$CO_2$ (black) simulations (a) shows the actual modelled global
mean surface air temperature (solid lines) whereas the dashed line shows the estimate based on
deep ocean temperatures, and (b) error in the estimate of global mean temperature change if based
on deep ocean temperatures (i.e. deep ocean – global mean surface temperatures).

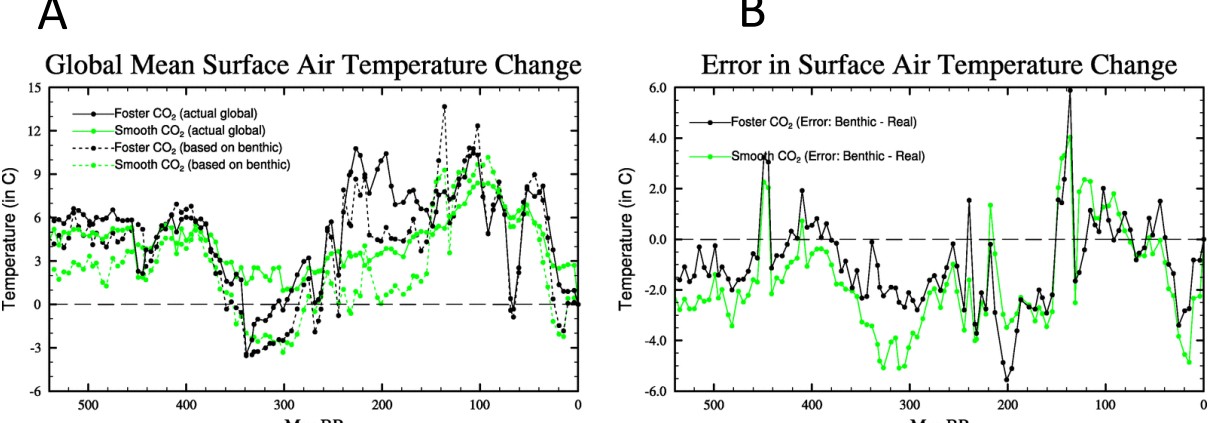

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

Table I. List of Paleogeographic Maps and PaleoDEMs

| Map Number | Stratigraphic Age Description | Plate Model Age |
|---|---|---|
| 1 | Present-day (Holocene, 0 Ma) | 0 |
| 2 | *Last Glacial Maximum (Pleistocene, 21 ky) * | 0 |
| 3 | *Late Pleistocene (122 ky) * | 0 |
| 4 | *Middle Pleistocene (454 ky) * | 0 |
| 5 | *Early Pleistocene (Calabrian, 1.29 Ma) * | 0 |
| 6 | *Early Pleistocene (Gelasian, 2.19) * | 0 |
| 7 | Late Pliocene (Piacenzian, 3.09) | 5 |
| 8 | *Early Pliocene (Zanclean, 4.47 Ma) * | 5 |
| 9 | *latest Miocene (Messinian, 6.3 Ma) * | 5 |
| 10 | Middle/Late Miocene (Serravallian&Tortonian, 10.5 Ma) | 10 |
| 11 | Middle Miocene (Langhian, 14.9 Ma) | 15 |
| 12 | Early Miocene (Aquitanian&Burdigalian, 19.5 Ma) | 20 |
| 13 | Late Oligocene (Chattian, 25.6 Ma) | 25 |
| 14 | Early Oligocene (Rupelian, 31 Ma) | 30 |
| 15 | Late Eocene (Priabonian, 35.9 Ma) | 35 |
| 16 | late Middle Eocene (Bartonian, 39.5 Ma) | 40 |
| 17 | early Middle Eocene (Lutetian, 44.5 Ma) | 45 |
| 18 | Early Eocene (Ypresian, 51.9 Ma) | 50 |
| 19 | Paleocene/Eocene Boundary (PETM, 56 Ma) | 55 |
| 20 | Paleocene (Danian&Thanetian, 61 Ma) | 60 |
| 21 | KT Boundary (latest Maastrichtian, 66 Ma) | 65 |
| 22 | Late Cretaceous (Maastrichtian, 69 Ma) | 70 |
| 23 | Late Cretaceous (Late Campanian, 75 Ma) | 75 |
| 24 | Late Cretaceous (Early Campanian, 80.8 Ma) | 80 |
| 25 | Late Cretaceous (Santonian&Coniacian, 86.7 Ma) | 85 |

| 82 | Late Devonian (early Famennian, 370 Ma) | 370 |
| 83 | Late Devonian (late Frasnian, 375 Ma) | 375 |
| 84 | Late Devonian (early Frasnian, 380 Ma) | 380 |
| 85 | Middle Devonian (Givetian, 385.2 Ma) | 385 |
| 86 | Middle Devonian (Eifelian, 390.5 Ma) | 390 |
| 87 | Early Devonian (late Emsian, 395 Ma) | 395 |
| 88 | Early Devonian (middle Emsian, 400 Ma) | 400 |
| 89 | Early Devonian (early Emsian, 405 Ma) | 405 |
| 90 | Early Devonian (Pragian, 409.2 Ma) | 410 |
| 91 | Early Devonian (Lochkovian, 415 Ma) | 415 |
| 92 | Late Silurian (Pridoli, 421.1 Ma) | 420 |
| 93 | Late Silurian (Ludlow, 425.2 Ma) | 425 |
| 94 | Middle Silurian (Wenlock, 430.4 Ma) | 430 |
| 95 | Early Silurian (late Llandovery, 436 Ma) | 435 |
| 96 | Early Silurian (early Llandovery, 441.2 Ma) | 440 |
| 97 | Late Ordovician (Hirnantian, 444.5 Ma) | 445 |
| 98 | Late Ordovician (Katian, 449.1 Ma) | 450 |
| 99 | Late Ordovician (Sandbian, 455.7 Ma) | 455 |
| 100 | Middle Ordovician (late Darwillian,460 Ma) | 460 |
| 101 | Middle Ordovician (early Darwillian,465 Ma) | 465 |
| 102 | Early Ordovician (Floian/Dapingianboundary, 470 Ma) | 470 |
| 103 | Early Ordovician (late Early Floian, 475 Ma) | 475 |
| 104 | Early Ordovician (Tremadoc, 481.6 Ma) | 480 |
| 105 | Cambro-Ordovician Boundary (485.4 Ma) | 485 |
| 106 | Late Cambrian (Jiangshanian, 491.8 Ma) | 490 |
| 107 | Late Cambrian (Pabian, 495.5 Ma) | 495 |
| 108 | late Middle Cambrian (Guzhangian, 498.8 Ma) | 500 |
| 109 | late Middle Cambrian (early Epoch 3, 505 Ma) | 505 |

| | | |
|---|---|---|
| 110 | early Middle Cambrian (late Epoch 2, 510 Ma) | 510 |
| 111 | early Middle Cambrian (middle Epoch 2, 515 Ma) | 515 |
| 112 | Early/Middle Cambrian boundary (520 Ma) | 520 |
| 113 | Early Cambrian (late Terreneuvian, 525 Ma) | 525 |
| 114 | Early Cambrian (middle Terreneuvian, 530 Ma) | 530 |
| 115 | Early Cambrian (early Terreneuvian, 535 Ma) | 535 |
| 116 | Cambrian/Precambrian boundary (541 Ma) | 540 |

*  *Simulations were not run for the time intervals highlighted in italics.*


Table 2. Summary of Model Simulations
The table summarises the simulations performed in this study. The 6th and 7th column refers to how
the model was initialised. The smooth $CO_2$ simulations were initialised from existing simulations
using paleogeographies which were provided commercial in confidence. The time period for which
these paleogeographies correspond to are listed in column 6, and the $CO_2$ value used in the runs is in
column 7. The Foster $CO_2$ simulations were initialised from the end point of the smooth $CO_2$
simulations.

| Time Period (in Ma) | $CO_2$ (in ppmv) for the Smooth $CO_2$ | $CO_2$ (in ppmv) for the Foster $CO_2$ | Length of Run (in years) (Smooth $CO_2$) | Length of Run (in years) (Foster $CO_2$) | Time period (in Ma) from existing simulations used for the initial condition for smooth CO2 simulation | $CO_2$ (in ppmv) used for the initial conditions |
|---|---|---|---|---|---|---|
| | | | | | | |
| 0.0 | 280 | 276 | 5000 | 5000 | 0 | 280 |
| 3.1 | 384 | 298 | 5000 | 2000 | 3 | 401 |
| 10.5 | 410 | 299 | 5000 | 2000 | 13 | 280 |
| 14.9 | 423 | 310 | 5000 | 2000 | 13 | 280 |
| 19.5 | 430 | 338 | 5000 | 2000 | 13 | 280 |
| 25.6 | 439 | 502 | 5000 | 2000 | 26 | 560 |
| 31.0 | 500 | 764 | 5000 | 5000 | 26 | 560 |
| 35.9 | 533 | 901 | 5000 | 5000 | 26 | 560 |
| 39.5 | 557 | 796 | 5000 | 5000 | 26 | 560 |
| 44.5 | 594 | 751 | 5000 | 8000 | 26 | 560 |
| 51.9 | 649 | 736 | 5000 | 8000 | 52 | 560 |

| | | | | | | |
|---|---|---|---|---|---|---|
| 56.0 | 630 | 570 | 5000 | 5000 | 52 | 560 |
| 61.0 | 604 | 335 | 5000 | 5000 | 52 | 560 |
| 66.0 | 576 | 229 | 5000 | 5000 | 69 | 560 |
| 69.0 | 560 | 262 | 5000 | 5000 | 69 | 560 |
| 75.0 | 633 | 559 | 5000 | 2000 | 69 | 560 |
| 80.8 | 704 | 667 | 5000 | 2000 | 69 | 560 |
| 86.7 | 775 | 590 | 5000 | 8000 | 69 | 560 |
| 91.9 | 839 | 466 | 5000 | 8000 | 92 | 560 |
| 97.2 | 840 | 707 | 5000 | 8000 | 92 | 560 |
| 102.6 | 840 | 1008 | 5000 | 5000 | 92 | 560 |
| 107.0 | 840 | 1028 | 5000 | 5000 | 92 | 560 |
| 111.0 | 827 | 1148 | 5000 | 8000 | 92 | 560 |
| 115.8 | 811 | 1103 | 5000 | 8000 | 92 | 560 |
| 121.8 | 784 | 986 | 5000 | 5000 | 92 | 560 |
| 127.2 | 752 | 898 | 5000 | 5000 | 92 | 560 |
| 131.2 | 728 | 896 | 5000 | 5000 | 92 | 560 |
| 136.4 | 699 | 1020 | 5000 | 8000 | 136 | 840 |
| 142.4 | 677 | 832 | 5000 | 8000 | 136 | 840 |
| 145.0 | 667 | 713 | 5000 | 8000 | 136 | 840 |
| 148.6 | 654 | 721 | 5000 | 8000 | 136 | 840 |
| 154.7 | 631 | 802 | 5000 | 5000 | 155 | 560 |
| 160.4 | 617 | 785 | 5000 | 5000 | 155 | 560 |
| 164.8 | 606 | 868 | 5000 | 8000 | 155 | 560 |
| 168.2 | 596 | 1019 | 5000 | 5000 | 167 | 840 |
| 172.2 | 581 | 1046 | 5000 | 5000 | 167 | 840 |
| 178.4 | 560 | 986 | 5000 | 5000 | 178 | 1120 |

| | | | | | | |
|---|---|---|---|---|---|---|
| 186.8 | 560 | 949 | 5000 | 5000 | 178 | 1120 |
| 190.8 | 560 | 1181 | 5000 | 5000 | 178 | 1120 |
| 196.0 | 560 | 1784 | 5000 | 5000 | 178 | 1120 |
| 201.3 | 560 | 1729 | 5000 | 5000 | 178 | 1120 |
| 204.9 | 560 | 1503 | 5000 | 5000 | 218 | 560 |
| 213.2 | 560 | 1223 | 5000 | 5000 | 218 | 560 |
| 217.8 | 560 | 1481 | 5000 | 5000 | 218 | 560 |
| 222.4 | 557 | 1810 | 5000 | 5000 | 218 | 560 |
| 227.0 | 553 | 2059 | 5000 | 5000 | 218 | 560 |
| 232.0 | 549 | 1614 | 5000 | 5000 | 218 | 560 |
| 233.6 | 548 | 1492 | 5000 | 5000 | 218 | 560 |
| 239.5 | 543 | 1034 | 5000 | 8000 | 218 | 560 |
| 244.6 | 540 | 419 | 5000 | 2000 | 218 | 560 |
| 252.0 | 534 | 879 | 5000 | 2000 | 257 | 1120 |
| 256.0 | 531 | 811 | 5000 | 2000 | 257 | 1120 |
| 262.5 | 526 | 352 | 5000 | 2000 | 257 | 1120 |
| 265.1 | 524 | 321 | 5000 | 2000 | 257 | 1120 |
| 268.7 | 521 | 311 | 5000 | 2000 | 257 | 1120 |
| 275.0 | 517 | 556 | 5000 | 2000 | 257 | 1120 |
| 280.0 | 513 | 690 | 5000 | 2000 | 257 | 1120 |
| 286.8 | 508 | 626 | 5000 | 2000 | 257 | 1120 |
| 292.6 | 503 | 495 | 5000 | 2000 | 297 | 280 |
| 297.0 | 500 | 445 | 5000 | 2000 | 297 | 280 |
| 301.3 | 510 | 393 | 5000 | 2000 | 297 | 280 |
| 305.4 | 520 | 358 | 5000 | 2000 | 297 | 280 |
| 311.1 | 534 | 338 | 5000 | 2000 | 297 | 280 |

| | | | | | | |
|---|---|---|---|---|---|---|
| 314.6 | 542 | 327 | 5000 | 2000 | 297 | 280 |
| 319.2 | 553 | 328 | 5000 | 2000 | 297 | 280 |
| 327.0 | 571 | 317 | 5000 | 2000 | 297 | 280 |
| 330.9 | 581 | 296 | 5000 | 2000 | 339 | 420 |
| 333.0 | 586 | 263 | 5000 | 2000 | 339 | 420 |
| 338.8 | 600 | 233 | 5000 | 2000 | 339 | 420 |
| 344.0 | 653 | 565 | 5000 | 2000 | 339 | 420 |
| 349.0 | 705 | 645 | 5000 | 2000 | 339 | 420 |
| 354.0 | 758 | 589 | 5000 | 2000 | 339 | 420 |
| 358.9 | 809 | 587 | 5000 | 2000 | 339 | 420 |
| 365.6 | 880 | 806 | 5000 | 2000 | 339 | 420 |
| 370.0 | 926 | 811 | 5000 | 2000 | 339 | 420 |
| 375.0 | 979 | 1052 | 5000 | 2000 | 339 | 420 |
| 380.0 | 1029 | 1269 | 5000 | 2000 | 339 | 420 |
| 385.2 | 1079 | 1377 | 5000 | 2000 | 377 | 1680 |
| 390.5 | 1131 | 1093 | 5000 | 2000 | 377 | 1680 |
| 395.0 | 1174 | 1297 | 5000 | 2000 | 377 | 1680 |
| 400.0 | 1223 | 1731 | 5000 | 2000 | 377 | 1680 |
| 405.0 | 1271 | 1689 | 5000 | 2000 | 377 | 1680 |
| 409.2 | 1319 | 2102 | 5000 | 2000 | 377 | 1680 |
| 415.0 | 1368 | 1579 | 5000 | 2000 | 377 | 1680 |
| 421.1 | 1427 | 1457 | 5000 | 2000 | 377 | 1680 |
| 425.2 | 1466 | 1490 | 5000 | 2000 | 377 | 1680 |
| 430.4 | 1517 | 1531 | 5000 | 2000 | 377 | 1680 |
| 436.0 | 1571 | 1576 | 5000 | 2000 | 377 | 1680 |
| 441.2 | 1614 | 1643 | 5000 | 2000 | 439 | 1877 |

| 444.5 | 1636 | 1708 | 5000 | 2000 | 439 | 1877 |
|-------|------|------|------|------|-----|------|
| 449.1 | 1666 | 1799 | 5000 | 2000 | 439 | 1877 |
| 455.7 | 1710 | 1929 | 5000 | 2000 | 439 | 1877 |
| 460.0 | 1738 | 2013 | 5000 | 2000 | 439 | 1877 |
| 465.0 | 1770 | 2111 | 5000 | 2000 | 439 | 1877 |
| 470.0 | 1803 | 2210 | 5000 | 2000 | 439 | 1877 |
| 475.0 | 1836 | 2308 | 5000 | 2000 | 439 | 1877 |
| 481.6 | 1879 | 2438 | 5000 | 2000 | 439 | 1877 |
| 485.4 | 1904 | 2513 | 5000 | 2000 | 439 | 1877 |
| 491.8 | 1946 | 2639 | 5000 | 2000 | 439 | 1877 |
| 495.5 | 1970 | 2711 | 5000 | 2000 | 439 | 1877 |
| 498.8 | 1992 | 2776 | 5000 | 2000 | 439 | 1877 |
| 505.0 | 2020 | 2870 | 5000 | 2000 | 439 | 1877 |
| 510.0 | 2040 | 2940 | 5000 | 2000 | 439 | 1877 |
| 515.0 | 2060 | 3010 | 5000 | 2000 | 439 | 1877 |
| 520.0 | 2080 | 3080 | 5000 | 2000 | 439 | 1877 |
| 525.0 | 2100 | 3150 | 5000 | 2000 | 439 | 1877 |
| 530.0 | 2120 | 3220 | 5000 | 2000 | 439 | 1877 |
| 535.0 | 2140 | 3290 | 5000 | 2000 | 439 | 1877 |
| 541.0 | 2164 | 3374 | 5000 | 2000 | 439 | 1877 |
