# Peer review of "Deep Ocean Temperatures through Time"

_Climate of the Past, 2020_

## Referee Comment (RC1) · Anonymous Referee #1 · 31 Jul 2020

TitleÂă: Deep Ocean Temperatures through Time

Authors: Valdes et al.

In this paper, the fundamental question of benthic oxygen isotopes as a proxy for global surface temperatures is addressed using a coupled General Circulation model (HadCM3). Based on extended set of simulations covering the whole Phanerozoic, this study demonstrates that the mean Earth's temperature is consistent with benthic oxygen isotopes for the Cenozoic era. Simulations using late Cretaceous boundary conditions suggest that deep ocean temperatures tend to be less well-correlated with polar temperatures. For deeper periods of time, the link becomes less precise and deep water temperatures cannot be interpreted anymore as a measure of the mean global mean temperature. If the CO2 is mentioned as a potential explanation, the

changing geography seems to be preferred.

Scientific interest

In my opinion, the present manuscript is good. The paper is very well written and considerable care has been taken to perform compare modeled temperatures and the proxy record (especially the statistical evaluation of the variability). Consequently this paper will be a useful resource for the community.

However this paper could become far better with an extended discussion. Here authors conclude that the change in the deep water temperature should not be taken as representative of the globally change of surface temperature for Mesozoic-Paleozoic eras. However the discrepancy between data and models is mainly driven by two warm periods (Jurassic and Triassic) where deep waters were formed in shallow sea at low latitudes (lines 517-519). This result is very interesting and challenges the initial suggestion of Emiliani (1954) that deep water continues to be formed at high latitudes in climate much warmer than today's. By adding a few diagnostics, the present study may become a key paper to understand the origin of warm deep waters and which conditions are required to form warm deep water at low latitudes (as initially proposed by Chamberlin, 1906). By identifying conditions for formation of deep waters in very warm climates (Cretaceous, Triassic), the scatter between deep ocean/mean surface temperatures should be significantly reduced for pre-Cenozoic periods. Since this result may substantially change the conclusion of the present paper, I recommend a major revision.

In addition to my general comment, here are some recommendations that the authors may consider to improve the paper (ranked by order of importance)

lines 381-395 (section Correlation of Deep Temperatures to Polar Sea Surface Temperatures) Here, the authors assume that most of the deep water in the model is formed at the surface at high latitudes (which explain why they explore the polar amplification later). However HadCM3 seems to be able to form dense warm saline bottom water in

subtropics when the Earth's climate becomes very warm. This process likely explains why purple dotes (fig. 6 and fig.9) are clearly above the proposed slope and why polar surface/deep ocean temperatures (fig.6) and deep ocean temperatures/global temperature (fig.9) are not well-correlated. Consequently this result seems to indicate that we need to distinguish two oceanic states for the formation of deep waters 1) when the main cause for buoyancy loss is salinity and 2) when the main cause for buoyancy loss is cooling.

This issue appears fundamental because this results implies that the change in polar surface temperature is not representative of the deep ocean temperatures. Consequently a "systematic" correlation between polar/deep ocean temperatures cannot be made. If the authors want to keep their initial conclusion, they have to demonstrate that the Emiliani's suggestion (1954) that most of deep waters continue to be formed at high latitudes is always true - even in a climate much warmer than today's.

For solving this issue, the authors should: - reconstruct the Earth temperature (for the whole Phanerozoic) using deep water temperatures assuming that "deep ocean water does not always form at polar latitudes" (line 386-387). - explore the effects of shallow sea at low latitudes for warm periods (here Triassic and Cretaceous) to decipher why the formation of deep water is so different, in both cases.

I realize that the authors may not want to undertake the project I have outlined. In that case they should claim only a speculation, not a conclusion, and they have to rewrite the paper in taking into account this major issue.

References to add: On a Possible Reversal of Deep-Sea Circulation and Its Influence on Geologic Climates. Chamberlin. T. Proceedings of the American Philosophical Society Vol. 45, No. 182 (1906), pp. 33-43 Temperatures of Pacific Bottom Waters and Polar Superficial Waters during the Tertiary, C. Emiliani Science (1954), Vol. 119, Issue 3103, pp. 853-855 DOI: 10.1126/science.119.3103.853

- lines 394-395: If Poulsen et al. (2001) has mentioned the formation of dense warm

saline water in subtropics by the geographic isolation (Mid-Cretaceous boundary conditions), Poulsen's simulations did not show a global circulation driven by deep water in the sub-tropics. Moreover, I do not understand why this result could be considered as model-dependant. Indeed both cases are generated by the same model.

- lines 349-355 (section 3.2: Comparison of Model Sea Surface Temperature to Proxy Data) I disagree with the following sentence - lines 351-352"... $15^\circ$C mismatch between models and data. If we assume the data has a seasonal bias, and select the summer seasons .... reduced by $4^\circ$C" This sentence implies that deep waters in polar regions are formed during the summer season (which means that the cause for buoyancy loss is salinity (without the formation of sea-ice)! Moreover this sentence is inconsistent with correlations made later in the paper. For example the caption of the figure 6 considers the polar temperature averaged in winter. This sentence should be removed and the discrepancy between models and data more discussed in the manuscript (see comment above).

- lines 508-513 (section Discussion and Conclusion) Paleogeographies are often mentioned as the main reason of the results outlined in the study. Unfortunately the direct effect of this factor is not well illustrated. Since the polar amplification depends on geographies used in this study, a figure showing the Poleward Energy Transport (or/and a paragraph) could be included in the section devoted to the " polar amplification"

- section 2.4 The authors implicitly assume that the initial state for the ocean has a marginal effect (which means that the final state is always the same whatever initial conditions used, so there is no hysteresis). However this point may have importance. Indeed, because late Jurassic deep waters are warmer than for the Cretaceous, a sensitivity test should be performed using temperature profiles obtained for Jurassic (instead of values from previous model simulations) to initialize the Cretaceous ocean.

- section 3.1 To demonstrate that the GCM is well designed to compare deep ocean temperatures to benthic ocean data, the revised version of the manuscript should contain a validation test using modern conditions (deep water temperatures simulated versus modern data) - or at least a reference.

- line 265 "less than 1000m" seems to be not consistent with the caption of the fig.4 (line 644)

- general shape of the manuscript The manuscript is organized by headings and subheadings from pages 2 to 10 but not after, why ?

---

## Referee Comment (RC2) · Anonymous Referee #2 · 11 Aug 2020

This paper explores the relationships between surface air temperature, polar temperature, and benthic temperature through geologic time using a series of HadCM3 simulations. Paleogeographies come from Scotese and Wright (2018), CO2 concentrations come from either Foster et al. (2017) or a new "smoothed" interpretation of CO2 records, solar evolution comes from Gough (1981), and orbital configuration is set to present-day. The number of simulations is very impressive. These data will be a valuable resource for the paleoclimate community. Regarding the manuscript, it is well written and easy to follow, but I have some recommendations before publication.

1) It would be useful to compare the proxy data with model data at the proxy collection sites instead of global averages. This could provide additional information about how well the current benthic records can reconstruct deep ocean temperature through time

and how well the benthic records reflect surface temperature.

2) Because the analyses largely focus on benthic temperature, I think benthic temperature is a better metric for determining equilibrium than global ocean temperature. Related, there is unexpected variability in some of the long-term temperature responses in figure 3. Can you rule out this type of response in the simulations that you only ran for 2000 years?

3) Additional details above how the "smoothed" $CO_2$ curve was derived are necessary. It sounds like it is based on temperature reconstructions, which makes its better agreement with the benthic records unsurprising. The information on lines 174-177 is not sufficient.

4) Why use a linear relationship for reconstructing surface temperature from deep ocean temperature? The relationship in figure 9 is non-linear. Might a non-linear fit improve the model based reconstructions?

5) How well does HadCM3 simulate present-day benthic temperature? If there are significant biases, maybe anomaly maps would make more sense. Also, a sentence about the dynamic vegetation would be nice.

Line 54 – Consider efforts using different models? (e.g. Donnadieu et al. (2016); Ladant et al. (2020))

Line 155 – Can we assume ocean salinity was constant through time?

Line 222 – What were the initial ocean conditions for the original simulation(s)?

Line 226 – All simulations were run in sequence? If not, please provide additional details about your spin up procedure.

Line 229 – Citation for the lack of multiple equilibria?

Line 230 – Impress number of simulations in near-equilibrium, but the values that define equilibrium seem a bit arbitrary.

Line 265 – Why only below 1000 m? Uncertainty in shelf reconstructions? Also, there is a discrepancy with a figure text stating below 300 m.

Line 266 – I don't fully understand the reasoning behind using 2000 m temperatures. Please explain further. Is it to suggest that other records from 2000 m can tell us about the global benthic temperature? Is it to suggest that the model does not need to reach deep ocean equilibrium?

Line 282 – You say paleosols and stomata underestimate CO2. Please provide citations. What CO2 reconstructions does this leave for deep time?

Line 291 – Many other citations for model-proxy discrepancy in the Miocene (e.g. Goldner et al., 2014)

Line 303 – Again, why use 2000 m temperature for comparison here?

Lines 309-313 – The proxy community would be very interested in a few more details on the evolution of deep water formation through time. Maybe a few plots of mixed layer depth would help show this?

Line 322 – Again, comparison at proxy locations would be more informative.

Line 352 – Sluggish circulation can also modify salinity and ocean d18O (e.g. Zhou et al., 2008)

Line 355 – Upchurch et al. (2015) also tested cloud microphysics.

Line 372 – I'm surprised that the mixed layer depth compared worse than polar temperature with benthic temperature.

Line 377 – Please show the response with just Cenozoic data

Line 395 – What happens to the high latitude temperature response when deep water forms primarily at the low latitudes?

Line 409 – Is HadCM3 underestimating the temperature gradient response or just underestimating sensitivity to CO2?

Line 434 – Can you speculate about what this means for proxy interpretation?

Line 440 – Isn't the albedo feedback somewhat offset by the fact that SSTs cannot get below 2°C?

Figures –

Need lettering on all panels. What each panel illustrates is not always clear.

Figure 1 – Land fraction is very low before 250 Ma! How does this impact your results?

Figure 2 – I think this may work better as a supplement figure with more time periods.

Figure 3 – What do the colors represent? Maybe easier to see with thinner lines.

Figure 4 – Should be "less than 1000 m". What is the Bemis latitudinal correction? Why not use a salinity based correction? Any correction for Mg/Ca?

Figure 5 – Some bias in the present-day simulation at 2000 m, which should be mentioned in the text.

Figure 6 – Are SSTs calculated below the sea ice? Might help to plot regressions for Foster and Smoothed separately.

Figure 7 – Reason for the large change in average ocean depth at ∼240 Ma?

Figure 10 – Dashed lines are a bit difficult to distinguish.

[Figure]

---

## Author Comment (AC1) · 13 Nov 2020

**Authors Response to review comments for:**

**Deep Ocean Temperatures through Time**

We thanks the reviewers for their comments, and have responded to all points. However, a few points are difficult to respond to within the context of these simulations. The simulations represent a huge amount of CPU time (approximately 1 million years of model simulation) requiring a significant fraction of a large HPC for about 9 months. It is therefore not practical to perform several further sets of simulations to investigate aspects, such as whether there are multiple equilibria. We therefore have had to amend our text to highlight the limitations of the work.

Similarly, we propose to produce a supplementary set of figures which are an atlas of the simulations. One page per time period (109 in total), showing the geographical maps for paleogeog, surface temperature, deep ocean temperature, mixed layer depth. These will address some of the requests from the reviewers.

**Response to Reviewer A**

(Grey text = reviewers comments, Black text= our response, Red text= addition/changes to paper)

**Major comment:**

"However this paper could become far better with an extended discussion. Here authors conclude that the change in the deep water temperature should not be taken as representative of the globally change of surface temperature for Mesozoic-Paleozoic eras.
However the discrepancy between data and models is mainly driven by two warm periods (Jurassic and Triassic) where deep waters were formed in shallow sea at low latitudes (lines 517-519). This result is very interesting and challenges the initial suggestion of Emiliani (1954) that deep water continues to be formed at high latitudes in climate much warmer than today's. By adding a few diagnostics, the present study may become a key paper to understand the origin of warm deep waters and which conditions are required to form warm deep water at low latitudes (as initially proposed by Chamberlin, 1906). By identifying conditions for formation of deep waters in very warm climates (Cretaceous, Triassic), the scatter between deep ocean/mean surface temperatures should be significantly reduced for pre-Cenozoic periods. Since this result may substantially change the conclusion of the present paper, I recommend a major revision."

We are not the first to suggest that there was deep water from the tropics. As discussed in lines 391 onwards, Brass et al 1982 was the first to suggest that the hypothesis of deep water always forming at high latitudes was not necessarily correct for all paleo times. Using a simple convective plume model, Brass et al showed "that the depth of termination and horizontal spreading of the plumes is directly related to their buoyancy flux which is the product of the density difference between the plume fluid and the surrounding environment and the volume flux of the plume. The plume with the greatest buoyancy flux will form the bottom water and plumes of lesser buoyancy flux will terminate and spread out at intermediate depths". Hence the key issue is where is the strongest

buoyancy flux. Brass et al then suggested that this could occur in warm conditions in the tropics, particularly if there was significant epicontinental seaways.

Similarly, we are not the first climate modelling group to show that this does occur.

The reviewer states that "…the discrepancy between data and models …". There is no discrepancy because, as far as I am aware, there is no data to show where deep-water is formed in the Jurassic and Triassic. Any discrepancy is between conceptual hypothesis (of Emiliani etc.) and the computer models presented here which follow the detailed laws of physics, such as Newton's laws and the laws of thermodynamics.

Thus none of our results and figures change depending on the interpretation (we cannot change the laws of physics). However, the reviewer does highlight our need to better explain our results, so we have expanded the paragraph discussing warm deep water, and included two extra figures. Specifically we have replaced the final text from line 391-394 by:

The idea that deep water may form in the tropics is in disagreement with early hypothesis (e.g. Emiliani (1954)) but has been previously suggested as a mechanism for warm Cretaceous deep water formation (Brass et al., 1982). Deep water typically forms in convective plumes. They showed that the depth and spreading of these plumes is related to the buoyancy flux with the greatest flux leading to bottom water and plumes of lesser flux leading to intermediate water. Brass et al (1982) suggested that this could occur in warm conditions in the tropics, particularly if there was significant epicontinental seaways and hypothesised that it "has been a dominant mechanism of deep water formation in historical times". It is caused by a strong buoyancy flux linked to strong evaporation at high temperatures.

Our computer model simulations are partly consistent with this hypothesis. The key aspect for the model is a relatively enclosed seaway in the tropics and warm conditions. The paleogeographic reconstructions (see supplementary figure 1) suggest an enclosed Tethyan-like seaway starting in the Carboniferous and extending through to the Jurassic and early Cretaceous. However, the colder conditions of the Carboniferous prevents strong tropical buoyancy fluxes. However, when we get into the Triassic and Jurassic, the warmer conditions lead to strong evaporation at low latitudes and bottom water formation in the tropics. This also explains why we see more tropical deep water (and hence poorer correlations between deep and polar surface temperatures in figure 6) when using the Foster et al $CO_2$ since this is generally higher (and hence warmer) than the smoothed record.

An example of the formation of tropical deep water is shown in figure xx. This shows a vertical cross-section of temperature and salinity near the equator for the Ladinian stage, mid-Triassic (240Ma). The salinity and temperature cross-section clearly shows high salinity warm waters sinking to the bottom of the ocean and spreading out. This is further confirmed by the water age tracer, figure yy. This shows the water age (measured as time since it experienced surface conditions, see England 1995) at 2731m in the model for the Permian, Triassic, Cretaceous and present day. The present day simulation shows that the youngest water is in the N. Atlantic and off the coast of Antarctica, indicating that this is where the deep water is forming. By contrast, the Triassic period shows that the youngest water is in the tropical Tethyan region and that it spreads out from there to fill the rest of the ocean basin. There is no young water at high latitudes, confirming that the source of bottom water is tropical only. For the Permian, although there continues to be a Tethyan-like tropical seaway, the colder conditions mean that deep water is again forming at high latitudes only. The Cretaceous is more complicated. It shows younger water in the high latitudes, but also shows some young water in the Tethys which merges with the high latitude waters. Additional indicator of the transitional nature of the Cretaceous are the mixed layer depth (see supplementary figure xx). This

is a measure of where water is mixing to deeper levels.  For this time period, there are regions of deep mixed layer in both the tropics and high latitudes, whereas it is only deep in the tropics for the Triassic and at high latitudes for present day.

This mechanism for warm deep water formation has also been seen in other climate models (e.g. [Barron and Peterson, 1990]). However, Poulsen et al 2001 conclude that in his model of the Cretaceous high-latitudes sources of deep water diminish with elevated $CO_2$ concentrations but did not see the dominance of tropical sources. Other models (e.g. Ladant et al 2020) do not show any significant tropical deep water formation, suggesting that this feature is potentially, a model-dependent result.

[Figure]

Figure xx. Longitudinal cross section at 20S of ocean potential temperature and salinity for the Ladanian (240Ma). Temperature is in C and salinity is in PSU.

[Figure]

Figure yy. Modelled age of water tracer at 2731m for 4 different time periods (a) 265Ma, (b) 240Ma, (c) 107Ma, and (d) 0Ma. Units are years.

**Minor Comments:**

lines 381-395 (section Correlation of Deep Temperatures to Polar Sea Surface Temperatures) Here, the authors assume that most of the deep water in the model is formed
at the surface at high latitudes (which explain why they explore the polar amplification
later). However HadCM3 seems to be able to form dense warm saline bottom water insubtropics
when the Earth's climate becomes very warm. This process likely explains why purple dotes (fig. 6
and fig.9) are clearly above the proposed slope and why polar surface/deep ocean temperatures
(fig.6) and deep ocean temperatures/global temperature (fig.9) are not well-correlated.

Consequently this result seems to indicate that we need to distinguish two oceanic states for the formation of deep waters 1) when the main cause for buoyancy loss is salinity and 2) when the main cause for buoyancy loss is cooling.

This issue appears fundamental because this results implies that the change in polar surface temperature is not representative of the deep ocean temperatures. Consequently a "systematic" correlation between polar/deep ocean temperatures cannot be made. If the authors want to keep their initial conclusion, they have to demonstrate that the Emiliani's suggestion (1954) that most of deep waters continue to be formed at high latitudes is always true - even in a climate much warmer than today's.
For solving this issue, the authors should: - reconstruct the Earth temperature (for the whole Phanerozoic) using deep water temperatures assuming that "deep ocean water does not always form at polar latitudes" (line 386-387). - explore the effects of shallow sea at low latitudes for warm periods (here Triassic and Cretaceous) to decipher why the formation of deep water is so different, in both cases.
I realize that the authors may not want to undertake the project I have outlined. In that case they should claim only a speculation, not a conclusion, and they have to rewrite the paper in taking into account this major issue.
References to add: On a Possible Reversal of Deep-Sea Circulation and Its Influence on Geologic Climates. Chamberlin. T. Proceedings of the American Philosophical Society Vol. 45, No. 182 (1906), pp. 33-43 Temperatures of Pacific Bottom Waters and Polar Superficial Waters during the Tertiary, C. Emiliani Science (1954), Vol. 119, Issue 3103, pp. 853-855 DOI: 10.1126/science.119.3103.853

This is a very similar comment to the major comment and we believe we have addressed the scientific issues in the previous response. The model does not assume anything about the location of deep water formation. This is a product of the laws of physics, as implemented within our model. We are plotting figure 6 to show that the "normal" assumption of the link between deep water and surface polar waters is not fully consistent by our physics-based model. We agree with the reviewer those time periods which show the largest departures are because of the deep water site, and we believe we have addressed the reasons for this in the revised text above.

We also note that the reviewer asks us to "reconstruct the Earth temperature (for the whole Phanerozoic) using deep water temperatures assuming that "deep ocean water does not always form at polar latitudes". This is impossible to directly address with a computer climate model. We cannot turn off/on where deep water forms. However, we do correlate the deep ocean water temperatures to global mean surface temperatures (see section starting at line 467 and figures 9 and 10). This confirms that there is a large time interval during the Paleozoic to the Jurassic when the errors at reconstructing global surface temperature from deep ocean temperatures are large.

- lines 394-395: If Poulsen et al. (2001) has mentioned the formation of dense warm saline water in subtropics by the geographic isolation (Mid-Cretaceous boundary conditions), Poulsen's simulations did not show a global circulation driven by deep water in the sub-tropics. Moreover, I do not understand why this result could be considered as model-dependant. Indeed both cases are generated by the same model.

The reviewer is correct that Poulsen did not show global changes driven by the tropics, but other models have. We have changed the text (see above).

- lines 349-355 (section 3.2: Comparison of Model Sea Surface Temperature to Proxy Data) I disagree with the following sentence - lines 351-352": : : 15_C mismatch between

models and data. If we assume the data has a seasonal bias, and select the summer seasons : : :. reduced by 4_C" This sentence implies that deep waters in polar regions are formed during the summer season (which means that the cause for buoyancy loss is salinity (without the formation of sea-ice)! Moreover this sentence is inconsistent with correlations made later in the paper. For example the caption of the figure 6 considers the polar temperature averaged in winter. This sentence should be removed and the discrepancy between models and data more discussed in the manuscript (see comment above).

In this section, we are comparing our surface temperatures to planktic foram reconstructions of SST, and are not discussing deep water data or mechanisms. The planktic data is potentially likely to be summer biased. We are not making any statement about deep water formation at this point. We have added a sentence to clarify this.

Of course, in practice deep water is formed during winter but the observed planktic data does not precisely record winter temperatures.

- lines 508-513 (section Discussion and Conclusion) Paleogeographies are often mentioned as the main reason of the results outlined in the study. Unfortunately the direct effect of this factor is not well illustrated. Since the polar amplification depends on geographies used in this study, a figure showing the Poleward Energy Transport (or/and a paragraph) could be included in the section devoted to the " polar amplification"

This is a really good point but highly non-trivial. The patterns of change in heat transport are complicated, and depend on numerous aspects of the paleogeography and other aspects of the model and we cannot identify any simple relationships. Whole papers have been written on changes in energy transport for one time period, let alone for 2 x 109 simulations presented here. A detaile analysis will be performed in a separate paper. For the moment, we have just highlighted the complexities in the following new paragraph, in the polar amplification section.

Changes in heat transport also play a potentially important role in polar amplification. Examination of the modelled poleward heat transport by the atmosphere and ocean shows a very complicated pattern, with all time periods showing the presence of some Bjerknes compensation (Bjerknes, 1964, and see, for example, Outten et al, 2018 for example in CMIP5 models). Bjerknes compensation is where the change in ocean transport is largely balanced by an equal but opposite change in atmospheric transport. For instance, compared to present day, the mid-Cretaceous and Early Eocene warm simulations shows a large increase in northward atmospheric heat transport, linked with enhanced latent heat transport associated with the warmer, moister atmosphere. However, this is partly cancelled by an equal but opposite change in the ocean transport. E.g. compared to present day, the early Eocene northern hemisphere atmospheric heat transport increases by up to 0.5PW, but the ocean   transport is reduced by an equal amount. The net transport from equator to the N.Pole changes by less than 0.1PW (i.e. less than 2% of total). Further back in time, the compensation is still apparent but the changes are more complicated, especially when the continents are largely  in the Southern hemisphere. Understanding the causes of these transport changes will be the subject of another paper.

- section 2.4 The authors implicitly assume that the initial state for the ocean has a marginal effect (which means that the final state is always the same whatever initial conditions used, so there is no hysteresis). However this point may have importance. Indeed, because late Jurassic deep waters are warmer than for the Cretaceous, a sensitivity test should be performed using temperature profiles obtained for Jurassic

(instead of values from previous model simulations) to initialize the Cretaceous ocean.

We explicitly acknowledge that our results could be dependent on the initial conditions (line 226). The existing simulations already account for a huge amount of computer time and it would be impractical to investigate the sensitivity to initial conditions for all time periods. This is because there is no systematic way to locate hysteresis without performing multiple simulations for each time period.  Simply initialising a Cretaceous model with Jurassic temperatures is not rigorous. We might show a transition, but we might not but the latter would not show that we do not have hysteresis, it would simply show that this particular sensitivity test did not show it. We have instead extended the discussion to expand the following caveat of our work:

Although it is always possible that a different initialization procedure may produce different final states, it is impossible to explore the possibility of hysteresis without performing many simulations for each period, which is currently beyond our computing resources. Previous studies using HadCM3L (not published) with alternative ocean initial states (isothermal at 0C, 8C, and 16C) have not revealed multiple equilibria but this might have been because we did not locate the appropriate part of parameter space that exhibits hysteresis. This remains a major caveat of our current work and which we wish to investigate, when we have sufficient computing resource.

- section 3.1 To demonstrate that the GCM is well designed to compare deep ocean temperatures to benthic ocean data, the revised version of the manuscript should contain a validation test using modern conditions (deep water temperatures simulated versus modern data) - or at least a reference.

This is shown in figure 5, where the final two images (d and e) are the modern model and Levitus observed data. However we did not describe this result. We have added a sentence to correct this.

Figure 5 also shows the modelled deep ocean temperatures for present day (Fig5d) compared to the World Ocean Atlas Data (fig5e). It can be seen that the broad patterns are well reproduced in the model, with good predictions of the mean temperature of the Pacific. The model is somewhat too warm in the Atlantic itself, and has a stronger plume from the Mediterranean than is shown in the observations.

- line 265 "less than 1000m" seems to be not consistent with the caption of the fig.4 (line 644)

The figure caption was incorrect.

- general shape of the manuscript The manuscript is organized by headings and subheadings from pages 2 to 10 but not after, why ?

Our mistake. Now corrected.

**Response to Reviewer 2**

1) It would be useful to compare the proxy data with model data at the proxy collection sites instead of global averages. This could provide additional information about how well the current benthic records can reconstruct deep ocean temperature through time and how well the benthic records reflect surface temperature.

We have added additional lines to show the benthic temperatures in Pacific and Atlantic basins separately.

We reiterate that the purpose of the comparison with data is to demonstrate that the model is broadly consistent with data but not a detailed validation. This is because the comparison is strongly influenced by the selection of CO2 and there is considerable uncertainty in this. We are currently undertaking a large 2-year project to perform a much more detailed comparison to data throughout the Phanerozoic, including the role of CO2.

2) Because the analyses largely focus on benthic temperature, I think benthic temperature is a better metric for determining equilibrium than global ocean temperature.

The metric we use is the volume integrated global ocean temperature and this is dominated by the slower time scales of the deep ocean and hence is a good metric, even for deep water. We have also examined temperatures averaged over the bottom 10 levels of the model, and temperatures at 2116m. They all show the same shape. We have added text at line 240.

The volume integrated temperature is dominated by the deep ocean so trends are similar if we look at the deep ocean only (e.g. at 2116m) or the average over the bottom 10 layers of the model.

Related, there is unexpected variability in some of the long-term temperature responses in figure 3. Can you rule out this type of response in the simulations that you only ran for 2000 years?

Those models that showed long-term behaviour (beyond 2000 years) failed our multiple criteria at year 2000, specifically although the trends might have been small, the energy balance was not. Hence we are reasonably confident that those models that passed all criteria at year 2000 were near equilibrium.

The strength of using multiple constraints is that a simulation may, by chance, pass one or two of these criteria but were unlikely to pass all three tests. For example, all of the models that we extended failed at least two of the criteria.

3) Additional details above how the "smoothed" CO2 curve was derived are necessary. It sounds like it is based on temperature reconstructions, which makes its better agreement with the benthic records unsurprising. The information on lines 174-177 is not sufficient.

We have added (from line 176):

Specifically, over the last decade and using proprietary paleogeographies, we performed multiple simulations at different $CO_2$ values for several stages across the last 440 million years and tested the resulting climate against proxy data (Harris et al., 2017). We then selected the $CO_2$ that best matched the data. For the current simulations, we linearly interpolated these $CO_2$ values for every stage. The resulting $CO_2$ curve happens to look like a heavily smoothed version of the Foster curve, but it was derived from a very different approach.

4) Why use a linear relationship for reconstructing surface temperature from deep ocean temperature? The relationship in figure 9 is non-linear. Might a non-linear fit improve the model based reconstructions?

It might improve the overall fit but there is relatively little justification for doing so. Except for the very coldest temperatures, the residuals of the linear fit are near Gaussian suggesting that there is no systematic evidence for non-linear variations. When studies use deep ocean temperatures to estimate global surface temperatures, they always assume linearity. We have added a line:

When we examine the residuals from the linear fit, they are near Gaussian suggesting that a linear fit is appropriate.

5) How well does HadCM3 simulate present-day benthic temperature? If there are significant biases, maybe anomaly maps would make more sense. Also, a sentence about the dynamic vegetation would be nice.

See comments in response to reviewer 1. We also point the reviewer to Valdes et 2017 which shows the vegetation, and have added at line 85:

The paper also documents the performance of the dynamic vegetation model but not the deep ocean temperatures. These are discussed in section 3.1

Line 54 – Consider efforts using different models? (e.g. Donnadieu et al. (2016); Ladant et al. (2020))

Added.

Line 155 – Can we assume ocean salinity was constant through time?

NO but we have little knowledge of how it might have changed. Added sentence.
We have little knowledge of whether ocean salinity has changed through time, but a global change in salinity has relatively little impact on the ocean circulation.

Line 222 – What were the initial ocean conditions for the original simulation(s)?

This is explained at line 232

Line 226 – All simulations were run in sequence? If not, please provide additional details about your spin up procedure.

NO. If we ran the models in sequence, the work would have taken more than 30 years to complete! We have added text at line 234:

Simulation were run in parallel so were not initialised from the previous stage results. In total, we performed almost 1 million years of model simulation and if we ran simulations in sequence, it would have taken 30 years to complete the simulations. By running these in parallel, initialised from previous modelling studies, we reduced the total run time to 3 months, albeit using a substantial amount of our high performance computer resources.

Line 229 – Citation for the lack of multiple equilibria?

None. If we had found multiple equilibria, we would have published! There are no papers because we have never found them.

Line 230 – Impress number of simulations in near-equilibrium, but the values that define equilibrium seem a bit arbitrary.

Agreed. The basis was that that trends and imbalances were smaller than typical orbital variability. Text added:

The numbers chosen were arbitrary but were to ensure that any trends and imbalances were smaller than typical trends associated with orbital variability.

Line 265 – Why only below 1000 m? Uncertainty in shelf reconstructions? Also, there is a discrepancy with a figure text stating below 300 m.

Figure caption corrected (see response to reviewer A). The choice of 1000m was also a bit arbitrary but was to avoid including shelf data which is generally not included in the compilations of benthic data. Text added at 265:

to avoid continental shelf locations which are typically not included in benthic data compilations .

Line 266 – I don't fully understand the reasoning behind using 2000 m temperatures. Please explain further. Is it to suggest that other records from 2000 m can tell us about the global benthic temperature? Is it to suggest that the model does not need to reach deep ocean equilibrium?

As mentioned in text, the data compilations can include data from ridges which are at 2000m in benthic compilations, and group all data together irrespective of depth (i.e. the data compilations essentially assume there are no vertical gradients between 2000m and the bottom of ocean). I selected the model level 2116m as a compromise. If I selected a lower level, e.g. at 4000m, then only a very small part of the ocean is sampled. If I used the bottom of the ocean, then it can be difficult to interpret since it includes a big range of depths. However, for all diagnostics, the difference between the bottom temperatures and the 2116m data were small. We have updated the text:

The observed data was collected from a range of depths - including mid-ocean ridges whose depth can vary from 2000m to the true bottom of the ocean. Thus benthic data compilations can be of variable depth and assume that the vertical temperature gradient is small.  And "…, and hence throughout the rest of the paper we frequently use the model 2116m temperatures as a surrogate for the true benthic temperature. This is also useful because the area of deep ocean can change substantially for even deeper levels."

Line 282 – You say paleosols and stomata underestimate CO2. Please provide citations. What CO2 reconstructions does this leave for deep time?

I have modified this line since it is difficult to find a reference (even though it is often discussed in workshops).

Both dips in the Foster-$CO_2$ simulations correspond to relatively low estimates of $CO_2$ concentrations. For these periods, the  dominant source of $CO_2$ values for these periods is from paleosols (fig.1) and thus we are reliant on one proxy methodology. At other time periods, the paleosols are often at the lower end of estimates (Foster et al 2017)

Line 291 – Many other citations for model-proxy discrepancy in the Miocene (e.g. Goldner et al., 2014)

I have added Goldner. I selected You et al because I think it was the first to show discrepancy. I was not attempting a comprehensive review.

Line 303 – Again, why use 2000 m temperature for comparison here?

See above.

Lines 309-313 – The proxy community would be very interested in a few more details on the evolution of deep water formation through time. Maybe a few plots of mixed layer depth would help show this?

Now included in supplementary information.

Line 322 – Again, comparison at proxy locations would be more informative.

See response to first comment.

Line 352 – Sluggish circulation can also modify salinity and ocean d18O (e.g. Zhou et al., 2008)

Modified text:

…, with weak horizontal temperature gradients at depth (though salinity gradients can still be important, Zhou et al 2008).

Line 355 – Upchurch et al. (2015) also tested cloud microphysics.

Reference added:

Line 372 – I'm surprised that the mixed layer depth compared worse than polar temperature with benthic temperature.

To be honest I was surprised too, but I think it is the complexity of the problem. The mixed layer depth is spatially, seasonally and duration varying in intensity and area. It was therefore difficult to precisely define the key sinking area, and to quantify the mass of water exchanged. A further study might attempt to do better quantification of the effects by outputting the mass exchange but we did not archive the variable in these simulations.

Line 377 – Please show the response with just Cenozoic data

The purpose of the paper is to model the whole of the Phanerozoic so we are reluctant to exclude other data. However, we have split figure 6 into all time periods (a), and Cenozoic  only (b)

Line 395 – What happens to the high latitude temperature response when deep water forms primarily at the low latitudes?

There is no obvious change to the response, but to fully answer such questions would require some additional sensitivity tests.

Line 409 – Is HadCM3 underestimating the temperature gradient response or just underestimating sensitivity to CO2?

The results of EoMIP (Lunt et al 2012) and DeepMIP (Lunt et al 2020) suggest that it is the temperature gradient is the biggest problem. I have added these references to the text.

Line 434 – Can you speculate about what this means for proxy interpretation?

Not sure what the reviewer is thinking about here, but have added:

The difference between the southern and northern hemisphere response complicates the interpretation of the proxies and leads to potentially substantial uncertainties. This is discussed in section 3.4 and figure 10.

Line 440 – Isn't the albedo feedback somewhat offset by the fact that SSTs cannot get below 2_C?

The reviewer is partly correct but the ice albedo back does dominate.

Figures –

Need lettering on all panels. What each panel illustrates is not always clear.

Done

Figure 1 – Land fraction is very low before 250 Ma! How does this impact your results?

This is partly discussed in Farnsworth et al 2019 and in the surface polar amplification section (now section 3.4)

Figure 2 – I think this may work better as a supplement figure with more time periods.

We have kept it but added supplementary figures with all time periods.

Figure 3 – What do the colors represent? Maybe easier to see with thinner lines.

Used thinner lines. Expanded figure caption to explain that each line represents a different time period.

Figure 4 – Should be "less than 1000 m". What is the Bemis latitudinal correction? Why not use a salinity based correction? Any correction for Mg/Ca?

Corrected. The data is from Cramwinkel and they performed all of the corrections. I have clarified this in the figure caption. Reference added for Bemis (correcting for latitudinal gradient of del18O of seawater).

Figure 5 – Some bias in the present-day simulation at 2000 m, which should be mentioned in the text.

Done. See response to reviewer A

Figure 6 – Are SSTs calculated below the sea ice? Might help to plot regressions for Foster and Smoothed separately.

Yes. Both lines added.

Figure 7 – Reason for the large change in average ocean depth at _240 Ma?

There is negligible ocean crust remaining from this time period so reconstructing the depth of the ocean is impossible.  It was arbitrarily set to 3680m depth. Added line to figure caption and to section 2.3 where we describe the paleogeogs.

Figure 10 – Dashed lines are a bit difficult to distinguish.

Improved

---

## Author Comment (AC2) · 13 Nov 2020

Our responses to both reviews are contained in the attached file.

---

## Author Response (AR1)

**Authors Response to review comments for:**

**Deep Ocean Temperatures through Time**

We thank the reviewers for their comments, and have responded to all points. However, a few of the more detailed points are difficult to respond to fully within the context of these simulations. Most published climate modelling papers tackle one time period, and maybe use one or a handful of model simulations. By contrast, we have 2 x 109 simulations using very different paleogeographies. Therefore it is impossible to go into the detail (e.g. why does the ocean heat transport change?) that a normal paper might target. To answer all such questions in depth would probably require 109 papers, one for each time period!

Similarly, the simulations represent a huge amount of CPU time (approximately 1 million years of model simulation) requiring a significant fraction of a large HPC for about 9 months. It is therefore not practical to perform several further sets of simulations to investigate some of the suggested additional aspects, such as whether there are multiple equilibria. We therefore have had to amend our text to highlight the limitations of the work rather than carry out additional simulations.

We have produced a supplementary set of figures which are an atlas of the simulations. One page per time period (109 in total), showing the geographical maps for paleogeography, surface temperature, deep ocean temperature, and mixed layer depth. This will also address some of the requests from the reviewers.

(Grey text = reviewers comments, Black text= our response, Red text= additions/changes to paper)

**Response to Editors Request**

The editor request we address the issue of the improved closuse of the water cyle in HadCM3, compared to previous simulations such as Farnsworth et al. (2019) and Lunt et al, 2016.

This is currently not possible to fully address because our modelling uses different paleogeographies and CO2 values. Farnsworth is currently re-running his set of simulations and will be preparing a paper which will address this issue. We have added a couple of sentences:

We have not directly compared our simulations to the previous runs of the (Farnsworth et al., 2019a) because they use different $CO_2$ and different paleogeographies. However in practice, the increase of salinity in their simulations is well mixed and seems to have relatively little impact on the overall climate and ocean circulation.

**Response to Reviewer A**

**Major comment:**

"However this paper could become far better with an extended discussion. Here authors conclude that the change in the deep water temperature should not be taken as representative

of the globally change of surface temperature for Mesozoic-Paleozoic eras.
However the discrepancy between data and models is mainly driven by two warm periods
(Jurassic and Triassic) where deep waters were formed in shallow sea at low
latitudes (lines 517-519). This result is very interesting and challenges the initial suggestion
of Emiliani (1954) that deep water continues to be formed at high latitudes in
climate much warmer than today's. By adding a few diagnostics, the present study may
become a key paper to understand the origin of warm deep waters and which conditions
are required to form warm deep water at low latitudes (as initially proposed by
Chamberlin, 1906). By identifying conditions for formation of deep waters in very warm
climates (Cretaceous, Triassic), the scatter between deep ocean/mean surface temperatures
should be significantly reduced for pre-Cenozoic periods. Since this result
may substantially change the conclusion of the present paper, I recommend a major
revision."

We are not the first to suggest that there was deep water formed in the tropics. As discussed in lines 391 onwards, Brass et al 1982 was the first to suggest that the hypothesis of deep water always forming at high latitudes was not necessarily correct for all paleo time periods. Using a simple convective plume model, Brass et al showed "that the depth of termination and horizontal spreading of the plumes is directly related to their buoyancy flux which is the product of the density difference between the plume fluid and the surrounding environment and the volume flux of the plume. The plume with the greatest buoyancy flux will form the bottom water and plumes of lesser buoyancy flux will terminate and spread out at intermediate depths". Hence the key issue is where is the strongest buoyancy flux. Brass et al then suggested that this could occur in warm conditions in the tropics, particularly if there was significant epicontinental seaways.

Similarly, we are not the first climate modelling group to show that this does occur in their model.

The reviewer states that "…the discrepancy between data and models …". There is no discrepancy because, as far as I am aware, there is no data to show where deep-water is formed in the Jurassic and Triassic. The reviewer discusses Emiliani but they only discuss data from the Tertiary and we do not have tropical deep water during these periods. Any discrepancy is due to an (invalid in our opinion) extrapolation of Emiliani to all geological times.

However, the reviewer does highlight our need to better explain our results, so we have expanded the paragraph discussing warm deep water, and included two extra figures. Specifically we have replaced the final text from line 391-394 by:

[revised manuscript text omitted]

Figure xx. Longitudinal cross section at 20S of ocean potential temperature and salinity for the Ladanian (240Ma). Temperature is in C and salinity is in PSU.

[Figure]

[Figure]

Figure yy. Modelled age of water tracer at 2731m for 4 different time periods (a) 265Ma, (b) 240Ma, (c) 107Ma, and (d) 0Ma. Units are years.

**Minor Comments:**

lines 381-395 (section Correlation of Deep Temperatures to Polar Sea Surface Temperatures) Here, the authors assume that most of the deep water in the model is formed
at the surface at high latitudes (which explain why they explore the polar amplification later). However HadCM3 seems to be able to form dense warm saline bottom water insubtropics when the Earth's climate becomes very warm. This process likely explains why purple dotes (fig. 6 and fig.9) are clearly above the proposed slope and why polar surface/deep ocean temperatures (fig.6) and deep ocean temperatures/global temperature (fig.9) are not well-correlated. Consequently this result seems to indicate that we need to distinguish two oceanic states for the formation of deep waters 1) when the main cause for buoyancy loss is salinity and 2) when the main cause for buoyancy loss is cooling.

This issue appears fundamental because this results implies that the change in polar surface temperature is not representative of the deep ocean temperatures. Consequently a "systematic" correlation between polar/deep ocean temperatures cannot be made. If the authors want to keep their initial conclusion, they have to demonstrate that the Emiliani's suggestion (1954) that most of deep waters continue to be formed at high latitudes is always true - even in a climate much warmer than today's.
For solving this issue, the authors should: - reconstruct the Earth temperature (for the whole Phanerozoic) using deep water temperatures assuming that "deep ocean water does not always form at polar latitudes" (line 386-387). - explore the effects of shallow sea at low latitudes for warm periods (here Triassic and Cretaceous) to decipher why the formation of deep water is so different, in both cases.
I realize that the authors may not want to undertake the project I have outlined. In that case they should claim only a speculation, not a conclusion, and they have to rewrite the paper in taking into account this major issue.
References to add: On a Possible Reversal of Deep-Sea Circulation and Its Influence on Geologic Climates. Chamberlin. T. Proceedings of the American Philosophical Society Vol. 45, No. 182 (1906), pp. 33-43 Temperatures of Pacific Bottom Waters and Polar Superficial Waters during the Tertiary, C. Emiliani Science (1954), Vol. 119, Issue 3103, pp. 853-855 DOI: 10.1126/science.119.3103.853

This is a very similar comment to the major comment and we believe we have addressed the scientific issues in the previous response. In particular, we want to highlight in the strongest terms

that the climate model does not assume anything about the location of deep water formation. It is a product of the laws of physics, as implemented within our model. The key thing is that when previous authors have used benthic temperatures as a proxy for global mean temperatures, it is they who assume the linkage with polar surface temperatures (as we discuss in the introduction and also see Hansen and Sato 2012 and many related papers).

We are plotting figure 6 to show that the "normal" assumption of the link between deep water and surface polar waters is not fully consistent with our physics-based model. We agree with the reviewer that those time periods which show the largest departures are because of the deep water site, and we believe we have addressed the reasons for this in the revised text above.

We also note that the reviewer asks us to "reconstruct the Earth temperature (for the whole Phanerozoic) using deep water temperatures assuming that "deep ocean water does not always form at polar latitudes". This is impossible to directly address with a computer climate model or indeed from data. We cannot turn off/on where deep water forms. However, we do correlate the deep ocean water temperatures to global mean surface temperatures (see section starting at line 467 and figures 9 and 10). This confirms that there is a large time interval during the Paleozoic to the Jurassic when the errors at reconstructing global surface temperature from deep ocean temperatures are large.

- lines 394-395: If Poulsen et al. (2001) has mentioned the formation of dense warm saline water in subtropics by the geographic isolation (Mid-Cretaceous boundary conditions), Poulsen's simulations did not show a global circulation driven by deep water in the sub-tropics. Moreover, I do not understand why this result could be considered as model-dependant. Indeed both cases are generated by the same model.

The reviewer is correct that Poulsen did not show global changes driven by the tropics, but other models have. We have changed the text (see above). We do not understand the final sentence - Poulsen uses a different climate model to our work.

- lines 349-355 (section 3.2: Comparison of Model Sea Surface Temperature to Proxy Data) I disagree with the following sentence - lines 351-352": : : 15_C mismatch between models and data. If we assume the data has a seasonal bias, and select the summer seasons : : :. reduced by 4_C" This sentence implies that deep waters in polar regions are formed during the summer season (which means that the cause for buoyancy loss is salinity (without the formation of sea-ice)! Moreover this sentence is inconsistent with correlations made later in the paper. For example the caption of the figure 6 considers the polar temperature averaged in winter. This sentence should be removed and the discrepancy between models and data more discussed in the manuscript (see comment above).

In this section, we are comparing our surface temperatures to planktic foram reconstructions of SST, and are not discussing deep water data or mechanisms. The planktic data has potential to be summer biased. We are not making any statement about deep water formation at this point. We have added a sentence to clarify this.

Of course, in practice deep water is formed during winter but the observed planktic data does not precisely record winter temperatures.

- lines 508-513 (section Discussion and Conclusion) Paleogeographies are often mentioned

as the main reason of the results outlined in the study. Unfortunately the direct
effect of this factor is not well illustrated. Since the polar amplification depends on geographies
used in this study, a figure showing the Poleward Energy Transport (or/and
a paragraph) could be included in the section devoted to the " polar amplification"

This is a really good point but highly non-trivial. The patterns of change in heat transport are complicated, and depend on numerous aspects of the paleogeography and other aspects of the model and we cannot identify any simple relationships. Whole papers have been written on changes in energy transport for one time period, let alone for 2 x 109 simulations presented here. A detailed analysis will be performed in a separate paper. For the moment, we have just highlighted the complexities in the following new paragraph, in the discussion section, and added a supplementary figure showing the complexity of transport.

Changes in heat transport also play a potentially important role in polar amplification. In the supplementary figure, we show the change in atmosphere and ocean poleward heat fluxes for each time period. Examination of the modelled poleward heat transport by the atmosphere and ocean shows a very complicated pattern, with all time periods showing the presence of some Bjerknes compensation (Bjerknes, 1964) (see (Outten et al., 2018) for example in CMIP5 models). Bjerknes compensation is where the change in ocean transport is largely balanced by an equal but opposite change in atmospheric transport. For instance, compared to present day, the mid-Cretaceous and Early Eocene warm simulations shows a large increase in northward atmospheric heat transport, linked with enhanced latent heat transport associated with the warmer, moister atmosphere. However, this is partly cancelled by an equal but opposite change in the ocean transport. E.g. compared to present day, the early Eocene northern hemisphere atmospheric heat transport increases by up to 0.5PW, but the ocean   transport is reduced by an equal amount. The net transport from equator to the N.Pole changes by less than 0.1PW (i.e. less than 2% of total). Further back in time, the compensation is still apparent but the changes are more complicated, especially when the continents are largely  in the Southern hemisphere. Understanding the causes of these transport changes will be the subject of another paper.

- section 2.4 The authors implicitly assume that the initial state for the ocean has a
marginal effect (which means that the final state is always the same whatever initial
conditions used, so there is no hysteresis). However this point may have importance.
Indeed, because late Jurassic deep waters are warmer than for the Cretaceous, a
sensitivity test should be performed using temperature profiles obtained for Jurassic
(instead of values from previous model simulations) to initialize the Cretaceous ocean.

We explicitly acknowledge that our results could be dependent on the initial conditions (line 226). The existing simulations already account for a huge amount of computer time and it would be impractical to investigate the sensitivity to initial conditions for all time periods. This is because there is no systematic way to locate hysteresis without performing multiple simulations for each time period.  Furthermore, it is impossible to prove the absence of hysteresis, only that we have not found it. We have instead extended the discussion to expand the following caveat of our work:

Although it is always possible that a different initialization procedure may produce different final states, it is impossible to explore the possibility of hysteresis or bistability without performing many simulations for each period, which is currently beyond our computing resources. Previous studies using HadCM3L (not published) with alternative ocean initial states (isothermal at 0C, 8C, and 16C) have not revealed multiple equilibria but this might have been because we did not locate the appropriate part of parameter space that exhibits bistability. However, other studies have shown such behaviour (e.g. (Baatsen et al., 2018)). This remains an avenue for further research, which we wish to investigate when we have sufficient computing resource.

- section 3.1 To demonstrate that the GCM is well designed to compare deep ocean temperatures to benthic ocean data, the revised version of the manuscript should contain a validation test using modern conditions (deep water temperatures simulated versus modern data) - or at least a reference.

This is shown in figure 5, where the final two images (d and e) are the modern model and Levitus observed data. However we did not describe this result. We have added a sentence to correct this.

Figure 5 also shows the modelled deep ocean temperatures for present day (Fig5d) compared to the World Ocean Atlas Data (fig5e). It can be seen that the broad patterns are well reproduced in the model, with good predictions of the mean temperature of the Pacific. The model is somewhat too warm in the Atlantic itself, and has a stronger plume from the Mediterranean than is shown in the observations.

- line 265 "less than 1000m" seems to be not consistent with the caption of the fig.4 (line 644)

The figure caption was incorrect.

- general shape of the manuscript The manuscript is organized by headings and subheadings from pages 2 to 10 but not after, why ?

Our mistake. Now corrected.

**Response to Reviewer 2**

1) It would be useful to compare the proxy data with model data at the proxy collection sites instead of global averages. This could provide additional information about how well the current benthic records can reconstruct deep ocean temperature through time and how well the benthic records reflect surface temperature.

We reiterate that the purpose of the comparison with data is to demonstrate that the model is broadly consistent with data but is not a detailed validation. This is because the comparison is strongly influenced by the selection of CO2 and there is considerable uncertainty in this. We are

currently undertaking a large 2-year project to perform a much more detailed comparison to point-based data throughout the Phanerozoic, including the role of CO2.

2) Because the analyses largely focus on benthic temperature, I think benthic temperature is a better metric for determining equilibrium than global ocean temperature.

Due to the colossal scale of our set of simulations (in total we have performed approximately 1 million years of simulation), we were unable to archive the full 3-D structure of the ocean and atmosphere for all years. We therefore have timeseries for selected variables and this does not include the benthic temperature. However, we argue that that the volume integrated global ocean temperature is dominated by the slower time scales of the deep ocean and hence is a good metric of ocean equilibrium, even for deep water. We have also examined the temperatures timeseries at 2731m (the deepest layer for which we have time series data) and it shows the same trends (see supplementary figures). We have added text at line 240.

We consider the volume integrated temperature because it includes all aspects of the ocean. However, it is dominated by the deep ocean trends and is near identical to the trends at 2731 (the lowest level that we have archived for the whole simulation).

Related, there is unexpected variability in some of the long-term temperature responses in figure 3. Can you rule out this type of response in the simulations that you only ran for 2000 years?

Those models that showed long-term behaviour (beyond 2000 years) failed our multiple criteria at year 2000, specifically although the trends might have been small, the energy balance was not. Hence we are reasonably confident that those models that passed all criteria at year 2000 were near equilibrium.  We also point out that even the shortest simulation of 2000 years is longer than PMIP protocols.

The strength of using multiple constraints is that a simulation may, by chance, pass one or two of these criteria but were unlikely to pass all three tests. For example, all of the models that we extended failed at least two of the criteria. The resulting time series of volume integrated global, annual mean ocean temperatures are shown in fig. 3. The supplementary figures also include this for each simulation, as well as the trends at 2731m.

3) Additional details above how the "smoothed" CO2 curve was derived are necessary. It sounds like it is based on temperature reconstructions, which makes its better agreement with the benthic records unsurprising. The information on lines 174-177 is not sufficient.

We have updated the text (from line 176) but have to also note that the full details are commercially confidential. In practice, it should be largely viewed simply as a what-if sensitivity test.

Secondly, the smoother curve was heavily influenced by a previous (commercially confidential) sparser sequence of simulations using non-public paleogeographic reconstructions. The resulting

simulations were generally in good agreement with terrestrial proxy datasets (Harris et al., 2017). Specifically, using proprietary (commercially confidential) paleogeographies, we have performed multiple simulations at different $CO_2$ values for several stages across the last 440 million years and tested the resulting climate against commercially confidential proxy data (Harris et al., 2017). We then selected the $CO_2$ that best matched the data. For the current simulations, we linearly interpolated these $CO_2$ values to every stage. The resulting $CO_2$ curve looks like a heavily smoothed version of the Foster curve and is within the (large) envelope of $CO_2$ reconstructions. The first-order shapes of the two curves are similar, though they are very different for some time periods (e.g. Triassic and Jurassic). In practice, both curves should be considered an approximation to the actual evolution of $CO_2$ through time, which remains uncertain.

4) Why use a linear relationship for reconstructing surface temperature from deep ocean temperature? The relationship in figure 9 is non-linear. Might a non-linear fit improve the model based reconstructions?

It might improve the overall fit but there is relatively little justification for doing so. Except for the very coldest temperatures, the residuals of the linear fit are near Gaussian suggesting that there is no systematic evidence for non-linear variations. Moreover when studies use deep ocean temperatures to estimate global surface temperatures, they always assume linearity. We have added a line:

Although we could have used a non-linear fit, when we examine the residuals from the linear fit, they are near Gaussian suggesting that a linear fit is appropriate analysis. Moreover, all previous use of benthic temperatures as a global mean surface temperature proxy are based on linear relationship, as far as we are aware.

5) How well does HadCM3 simulate present-day benthic temperature? If there are significant biases, maybe anomaly maps would make more sense. Also, a sentence about the dynamic vegetation would be nice.

See comments in response to reviewer 1. We also point the reviewer to Valdes et 2017 which shows the vegetation, and have added at line 85:

The performance of the dynamic vegetation model is also described in (Valdes et al., 2017) but the deep ocean temperatures are not described in that paper. We therefore include a comparison to present day observed deep ocean temperatures in section 3.1

Line 54 – Consider efforts using different models? (e.g. Donnadieu et al. (2016); Ladant et al. (2020))

Added.

Line 155 – Can we assume ocean salinity was constant through time?

No, but we have little knowledge of how it might have changed. Added sentence.
We have little knowledge of whether ocean salinity has changed through time, but a global mean change in salinity has relatively little impact on the ocean circulation due to the near-linearity of the equation of state over much of the range of modelled salinities and temperatures.

Line 222 – What were the initial ocean conditions for the original simulation(s)?

This is explained at line 232

Line 226 – All simulations were run in sequence? If not, please provide additional details about your spin up procedure.

No, if we ran the models in sequence, the work would have taken more than 30 years to complete! We have added text at line 234:

Simulations were run in parallel so were not initialised from the previous stage results. In total, we performed almost 1 million years of model simulation and if we ran simulations in sequence, it would have taken 30 years to complete the simulations. By running these in parallel, initialised from previous modelling studies, we reduced the total run time to 3 months, albeit using a substantial amount of our high performance computer resources.

Line 229 – Citation for the lack of multiple equilibria?

None. If we had found multiple equilibria, we would have published! There are no papers because we have never found them.

Line 230 – Impress number of simulations in near-equilibrium, but the values that define equilibrium seem a bit arbitrary.

Agreed. The basis was that that trends and imbalances were smaller than typical orbital variability. Text added:

The numbers chosen were arbitrary but were to ensure that any trends and imbalances were smaller than typical trends associated with orbital variability.

Line 265 – Why only below 1000 m? Uncertainty in shelf reconstructions? Also, there is a discrepancy with a figure text stating below 300 m.

Figure caption corrected (see response to reviewer A). The choice of 1000m was also a bit arbitrary but was to avoid including shelf data which is generally not included in the compilations of benthic data. Text added at 265:

to avoid continental shelf locations which are typically not included in benthic data compilations .

Line 266 – I don't fully understand the reasoning behind using 2000 m temperatures. Please explain further. Is it to suggest that other records from 2000 m can tell us about the global benthic temperature? Is it to suggest that the model does not need to reach deep ocean equilibrium?

As mentioned in text, the data compilations can include data from ridges which are at 2000m in benthic compilations, and group all data together irrespective of depth (i.e. the data compilations essentially assume there are no vertical gradients between 2000m and the bottom of ocean). I selected the model level 2116m as a compromise. If I selected a lower level, e.g. at 4000m, then only a very small part of the ocean is sampled. If I used the bottom of the ocean, then it can be difficult to interpret since it includes a big range of depths and hence there are strong gradients purely because of the depth variations. However, following the comment of the reviewer I have increased the depth to 2731m as it probably is more representative of the average from cores. It would be great for a data person to go over the existing compilations to better reflect the depth of the cores, but this would be a massive effort since the compilations have ~10000 data points and currently do not always record depth. However, for all diagnostics, it should be noted that the difference between the bottom temperatures and the 2116 or 2731m data were small. We have updated the text:

The observed benthic data are collected from a range of depths and are rarely at the very deepest levels (e.g. the new cores in (Friedrich et al., 2011) range from current water depths ranging from 1899m to 3192m). Furthermore large data compilations rarely include the depth profiles and thus effectively assume that any differences between basins and through time are entirely due to climate change and not to changes in depth. Hence throughout the rest of the paper we frequently use the modelled 2731m temperatures as a surrogate for the true benthic temperature. This is a pragmatic definition because the area of deep ocean reduces rapidly (e.g. there is typically only 50% of the globe deeper than 3300m). To evaluate whether this procedure gives a reasonable result, we also calculated the global average temperature at model layer 2731m. This is shown by the dashed line in figure 4a. In general, the agreement between model bottom water temperatures and 2731km temperatures is very good. The standard deviation is 0.7˚C, and the maximum difference is 1.4˚C. Compared to the overall variability, this is a relatively small difference and shows that it is reasonable to assume that the deep ocean has weak vertical gradients.

Line 282 – You say paleosols and stomata underestimate CO2. Please provide citations.
What CO2 reconstructions does this leave for deep time?

I have modified this line. It probably isn't correct for paleosols and is difficult to find a reference for stomata (even though it is often discussed in workshops).

For these periods, the dominant source of $CO_2$ values for these periods is from paleosols (fig.1) and thus we are reliant on one proxy methodology. Unfortunately, the alternative $CO_2$ reconstructions of (Witkowski et al., 2018) have a data gap during this period.

Line 291 – Many other citations for model-proxy discrepancy in the Miocene (e.g. Goldner et al., 2014)

I have added Goldner (and also Krapp and Knorr). I selected You et al because I think it was the first to show discrepancy. I was not attempting a comprehensive review of the Miocene modelling.

Line 303 – Again, why use 2000 m temperature for comparison here?

See above.

Lines 309-313 – The proxy community would be very interested in a few more details on the evolution of deep water formation through time. Maybe a few plots of mixed layer depth would help show this?

Now included in supplementary information.

Line 322 – Again, comparison at proxy locations would be more informative.

See response to first comment.

Line 352 – Sluggish circulation can also modify salinity and ocean d18O (e.g. Zhou et al., 2008)
Modified text:
…, with weak horizontal temperature gradients at depth (though salinity gradients can still be important, Zhou et al 2008).

Line 355 – Upchurch et al. (2015) also tested cloud microphysics.

Reference added:
Line 372 – I'm surprised that the mixed layer depth compared worse than polar temperature with benthic temperature.

To be honest we were surprised too, but I think it is the complexity of the problem. The mixed layer depth varies (spatially, seasonally and in duration) in intensity and area. It was therefore difficult to precisely define the key sinking area, and to quantify the mass of water exchanged. A further study might attempt to do better quantification of the effects by outputting the mass exchange but we did not archive the variable in these simulations.

Line 377 – Please show the response with just Cenozoic data

The purpose of the paper is to model the whole of the Phanerozoic so we are reluctant to exclude other data. Figure 6 already shows the different time periods in different colours.

Line 395 – What happens to the high latitude temperature response when deep water forms primarily at the low latitudes?

If you mean "how does the high latitude respond to changing CO2 etc", then there is no obvious change to the response, but to fully answer such questions would require some additional sensitivity tests.

Line 409 – Is HadCM3 underestimating the temperature gradient response or just underestimating sensitivity to CO2?

The results of EoMIP (Lunt et al 2012) and DeepMIP (Lunt et al 2020) suggest that it is the temperature gradient is the biggest problem. I have added these references to the text.

Line 434 – Can you speculate about what this means for proxy interpretation?

Not sure what the reviewer is thinking about here, but have added:

The difference between the southern and northern hemisphere response complicates the interpretation of the proxies and leads to potentially substantial uncertainties. This is discussed in section 3.4 and shown in figure 10.

Line 440 – Isn't the albedo feedback somewhat offset by the fact that SSTs cannot get below 2_C?

The reviewer is partly correct, but without carefully controlled sensitivity simulations, it is difficult to quantify the relative importance.

Figures –

NOTE that the current figures are gif files but all are available with higher quality pdf files.

Need lettering on all panels. What each panel illustrates is not always clear.

Done

Figure 1 – Land fraction is very low before 250 Ma! How does this impact your results?

This is discussed in Farnsworth et al 2019 and in the surface polar amplification section (now section 3.4). We don't have any further insights.

Figure 2 – I think this may work better as a supplement figure with more time periods.

We have kept it but added supplementary figures with all time periods.

Figure 3 – What do the colors represent? Maybe easier to see with thinner lines.

Used thinner lines. Expanded figure caption to explain that each line represents a different time period. Also added for each simulation in the supplementary text.

Figure 4 – Should be "less than 1000 m". What is the Bemis latitudinal correction? Why not use a salinity based correction? Any correction for Mg/Ca?

Corrected. The data is from Cramwinkel and they performed all of the corrections. I have therefore removed this and pointed the reader to the Cramwinkel paper.

Figure 5 – Some bias in the present-day simulation at 2000 m, which should be mentioned in the text.

Done. See response to reviewer A

Figure 6 – Are SSTs calculated below the sea ice? Might help to plot regressions for Foster and Smoothed separately.

Yes. SSTs will be -2C below ice. Both lines added.

Figure 7 – Reason for the large change in average ocean depth at _240 Ma?

There is negligible ocean crust remaining from this time period so reconstructing the depth of the ocean is impossible.  It was arbitrarily set to 3680m depth. Added line to figure caption and to section 2.3 where we describe the paleogeographies.

Figure 10 – Dashed lines are a bit difficult to distinguish.

Improved

---

## Author Response (AR2)

**Authors Response to review comments for:**

**Deep Ocean Temperatures through Time**

We thank the reviewer for their additional comments, and have responded to all points. Grey text are the referee comments, black text is our response, and red text is the text added to manuscript.

"We have also examined the temperatures timeseries at 2731m (the deepest layer for which we have time series data) and it shows the same trends (see supplementary figures)."
The supplemental figures are very helpful for understand the responses. I agree that global and 2731m ocean temperatures generally show the same trends. However, global and 2731m ocean temperature do not have a consistent offset between time periods, which is very interesting and important for comparing models with proxies. Hopefully these differences will be explored in great detail in future works.

Yes. We will almost certainly will do. The changes in vertical gradients in the ocean are intriguing.

By the way, there are some formatting issues and inconsistencies in the supplemental figures. Please fix before publication.

Done.

"Except for the very coldest temperatures, the residuals of the linear fit are near Gaussian suggesting that there is no systematic evidence for non-linear variations. Moreover when studies use deep ocean temperatures to estimate global surface temperatures, they always assume linearity."
That's my point. You would not expect a linear relationship at low temperatures. I thought previous works distinguished between warm and cold climates.

In some recent papers, they may have done this but in the original papers by Hansen etc. they used a simple linear fit. We have added a further sentence to emphasis this.

This suggests that using a simple linear relationship (as in (Hansen et al., 2008)) could be improved upon.

The ocean initial conditions are still not clear. "we initialized the ocean temperatures and salinity with the values from previous model simulations from similar time periods" does not say much. From the supplemental, some simulations start cold, and some simulations start warm. Is initial salinity the same for all simulations? Are the initial conditions to blame for the unusual jumps in ocean temperature in some simulations (e.g. preindustrial)?

We have added further sentences explaining the initial conditions. We have also added a new table (2) which shows the length of all simulations, and the initial conditions. The unusual jumps at the start of the simulation were an attempt to speed-up convergence by linearly interpolating the trends. We have also explained this.

To speed up the convergence of the model, we initialized the ocean temperatures and salinity with the values from previous model simulations from similar time periods using the commercial in confidence paleogeographies. Specifically, we had a set of 17 simulations covering the last 440Ma. We selected the nearest simulation to the time period. For instance, the 10.5 Ma, 14.9 Ma, and the

19.5Ma simulations were initialised from the 13Ma simulation performed using the alternative paleogeographies. Table 2 summarises the simulations performed in this study and shows the initialisation of the model. The Foster $CO_2$ simulations were initialised from the end point of the smooth $CO_2$ simulations. In the first set of simulations (smooth $CO_2$) we also attempted to accelerate the spin up by using the ocean temperature trends at year 500 to linearly extrapolate the bottom 10 level temperatures for a further 1000 years. This had limited success and was not repeated. The atmosphere variables were also initialized from the previous model simulations but the spin-up of the atmosphere is much more rapid and did not require further intervention.

I thought HadCM3 requires a piece of land at the poles, is this not the case? It does not appear in the supplemental land mask figures.

HadCM3 has to have a polar island in the ocean model but the atmosphere model can handle the correct surface type. For instance, in the present day model there is land at the singularity at the N.Pole but in the atmosphere model it "sees" seaice. Therefore the principle issue is that the ocean model does not allow flow across the pole. The polar stereographic plots in the supplementary is showing the atmosphere grid (because it is showing whether there is landice). The other plots are Mollweide and the ocean plots do have land at the poles. However, because Mollweide is an area conserving projection, it is impossible to see the polar islands. We have added text to explain this.

To avoid singularity at the poles, the ocean model always has to have land at the poles (90N and 90S), but the atmosphere model can represent the poles correctly (i.e. in the pre-industrial geography, the atmosphere considers there is sea ice covered ocean at the N.Pole but the ocean model has land and hence there is no ocean flow across the pole).

We also took the opportunity to correct some spelling errors and improve the grammar.